# SLIM: Secure and Efficient Inference for Large Language Models on Untrusted Devices via TEEs

Wei Wang [* 1]  Zihao Guan [* 1]  Xing Zhou [2]  Yan Ding [1]  Yusong Tan [1]  Jie Yu [1]  Bao Li [1]

## Abstract

Deploying large language models (LLMs) on untrusted hardware entails a risk of weight extraction, which can lead to unauthorized replication and misuse of the model. A practical approach is to leverage Trusted Execution Environments (TEEs) and protect model security by obfuscating model weights. However, existing obfuscation schemes struggle to simultaneously provide strong security guarantees and high performance: schemes with security guarantees incur substantial overhead due to frequent TEE interactions, whereas schemes that achieve efficient inference are insecure. We propose SLIM, a secure inference framework that exploits the iterative structure of LLMs to let transformed representations cascade through consecutive obfuscated layers, thereby minimizing interactions with the TEE. SLIM introduces a T-Way Mixing algorithm that performs consecutive inter-vector covering using carefully constructed block-diagonal Householder matrices and combines it with successive random permutations, providing thorough weight obfuscation while keeping TEE-side computation lightweight. Evaluations demonstrate that SLIM provides robust security guarantees and significantly outperforms prior state-of-the-art obfuscation schemes in terms of performance, delivering up to a $13.80\times$ speedup while preserving fidelity.

## 1. Introduction

As deployment paradigms for large language models (LLMs) become increasingly diverse, models are now commonly hosted across multiple administrative domains. In practice, LLM providers often deploy proprietary models on edge devices, while model developers typically favor cloud-based deployments. This fragmented, multi-end deployment landscape significantly enlarges the attack surface for model extraction, making model theft a critical obstacle to the secure and reliable adoption of LLMs. When these LLMs are deployed on untrusted devices and lack adequate security protections, attackers can employ software analysis techniques to extract the deployed model, including architectures and weights, thereby compromising the intellectual property of the LLM (Sun et al., 2021). Therefore, robust protection mechanisms for LLMs are of paramount importance, given the substantial costs and private data required to develop high-performance LLMs (Sharir et al., 2020).

Trusted Execution Environments (TEEs) are widely deployed hardware-isolation technologies across a broad range of devices and have been shown in prior work to provide a practical solution for protecting model weights on untrusted devices (Wang et al., 2025; Zhang et al., 2024b). Given the limited computational throughput and enclave resources of TEEs, prior work commonly obfuscates compute-intensive layers and offloads their execution to an untrusted GPU, while restricting the TEE to obfuscation and de-obfuscation (recovery). This division of labor improves inference efficiency while preserving model confidentiality. However, modern LLMs (e.g., LLaMA (Dubey et al., 2024) and DeepSeek-R1 (Guo et al., 2025)) comprise tens to hundreds of consecutive layers, and existing secure obfuscation mechanisms typically interact with the TEE on a per-layer basis, resulting in substantial inference overhead.

Prior TEE-based obfuscation schemes (Shen et al., 2022; Sun et al., 2023; Li et al., 2024; 2025) obfuscate weights in a per-vector manner via operations such as scaling and row/column permutations. However, these operations cannot disrupt vector directions: GroupCover (Zhang et al., 2024b) demonstrates that the above schemes are vulnerable to model-stealing attacks based on statistical analysis. To strengthen security, GroupCover defends via inter-vector coverage, whereas ArrowCloak (Wang et al., 2025) further reduces the per-invocation TEE computation cost by leveraging matrix–vector multiplication. However, both approaches still require per-layer interaction with the TEE, leading to

---
*Equal contribution [1]College of Computer Science and Technology, National University of Defense Technology, Changsha, China [2]College of Intelligence Science and Technology, National University of Defense Technology, Changsha, China. Correspondence to: Yan Ding <yanding@nudt.edu.cn>, Yusong Tan <ystan@nudt.edu.cn>.

*Proceedings of the $43^{rd}$ International Conference on Machine Learning*, Seoul, South Korea. PMLR 306, 2026. Copyright 2026 by the author(s).

substantial TEE-GPU transfer overhead. Therefore, ensuring secure inference for LLMs while avoiding prohibitive performance overhead remains an open challenge.

To address the above challenge, we propose a **S**ecure **L**ightweight **I**nference framework for **L**LMs, termed SLIM[1]. Guided by the structural features of LLM architectures, SLIM enables the TEE-transformed data to cascade and to progressively propagate through obfuscated network layers, ultimately yielding correct inference results. To ensure both strong security and lightweight execution, we design the *T-Way Mixing Algorithm*, which performs successive vector-covering operations based on the carefully constructed Householder matrices. This design guarantees the effectiveness of weight obfuscation while leveraging vectorized operations to reduce the computational complexity inside the TEE by an order of magnitude. Through rigorous theoretical analysis and extensive experimental evaluation, we demonstrate that SLIM achieves both strong security guarantees and high inference efficiency.

Our main contributions are summarized as follows:

- We propose SLIM, a propagation-compatible TEE-based secure inference framework for LLMs that improves prior propagation protocols by replacing direction-preserving transformations with direction-hiding orthogonal obfuscation.

- We introduce a lightweight obfuscation algorithm, termed the *T-Way Mixing Algorithm*, which achieves effective obfuscation of weights while ensuring that the computation performed inside the resource-constrained TEE is limited to lightweight matrix–vector multiplications.

- Extensive theoretical analysis and empirical evaluations demonstrate that SLIM can defend against the most advanced attacks. SLIM significantly outperforms existing state-of-the-art schemes, achieving up to a $13.80\times$ reduction in the time to first token (TTFT) and up to a $4.23\times$ increase in throughput, while incurring negligible inference accuracy loss.

**Conflict of Interest Disclosure.** The authors have no financial conflicts of interest to declare.

## 2. Related Work

**TEE-based Model-Isolated Inference.** This approach isolates the model inference process from the external environment using a TEE to prevent theft or tampering of inference data or model weights. Schemes of this type encrypt

---

[1] The code is available at: https://github.com/nuanyang-wei/SLIM

*Table 1.* Comparison with prior work. Data Trans. and TEE Comp. refer to the TEE-GPU data transfer overhead and TEE computation overhead, respectively. $d$ as the hidden size, $n$ as the number of Decoder Blocks, $l$ as the input token length, $k$ as the block size of the transformation matrix. Legend: ◯= not supported, ◑= partially supported, ●= fully supported. For a more detailed comparison, please refer to the Appendix A.

| Scheme | Data Trans. | TEE Comp. | LLM Compatibility | Security |
|---|---|---|---|---|
| SOTER | $O(nld)$ | $O(nld)$ | ◑ | ◯ |
| ShadowNet | $O(nld)$ | $O(nld)$ | ◑ | ◯ |
| GroupCover | $O(nld)$ | $O(nldk)$ | ◑ | ● |
| TransLinkGuard | $O(nld)$ | $O(nld)$ | ● | ◯ |
| CoreGuard | $O(ld)$ | $O(ld)$ | ● | ◯ |
| TSQP | $O(nld)$ | $O(nld)$ | ◑ | ◯ |
| ArrowCloak | $O(nld)$ | $O(nld)$ | ◑ | ● |
| SLIM (Ours) | $O(ld)$ | $O(ld\log_k d)$ | ● | ● |

the model and decrypt it for computation within the TEE. Works such as Confidential DL (VanNostrand et al., 2019), Real-Time DNN (Babar & Hasan, 2022), and T-Slices (Islam et al., 2023) ensure attackers cannot obtain the model weights by partitioning the full model into layers, loading and decrypting each layer sequentially for execution within the TEE. However, a significant drawback of this approach is its reliance solely on TEE computation, which cannot leverage GPU computing resources, making it difficult to apply to LLM inference. To overcome the resource and computational limitations of TEEs, some works opt to isolate and protect only critical layers (Elgamal & Nahrstedt, 2020; Xiang et al., 2021; Mo et al., 2020). A key flaw in such schemes, however, is **the direct exposure of a significant number of model weights from the unprotected layers, making it challenging to guarantee security**.

**TEE-based Obfuscated Inference.** TEE-based obfuscated inference adopts a variety of weight obfuscation techniques to protect the weight values of the model. To reduce TEE-side computation, Magnitude (Hou et al., 2022) and NNSplitter (Zhou et al., 2023) obfuscate only a small subset of critical weights; however, such targeted modifications induce anomalous weight distributions that can be quickly detected and reversed. SOTER (Shen et al., 2022), ShadowNet (Sun et al., 2023), TSQP (Sun et al., 2025), TransLinkGuard (Li et al., 2024), and CoreGuard (Li et al., 2025) adopt vector scaling or channel shuffling to reduce recovery cost, yet they do not protect the directional properties of private vectors and have been shown to **be vulnerable to statistical-analysis-based attacks** (Zhang et al., 2024b; Wang et al., 2025). TEE-Slice (Zhang et al., 2024a) protects models by training a subset of layers inside the TEE, but this constrains the training regime. GroupCover (Zhang et al., 2024b) and ArrowCloak (Wang et al., 2025) disrupt the directional distribution of model weights via inter-vector coverage operations. However, these approaches require per-layer interaction with the TEE, incurring significant

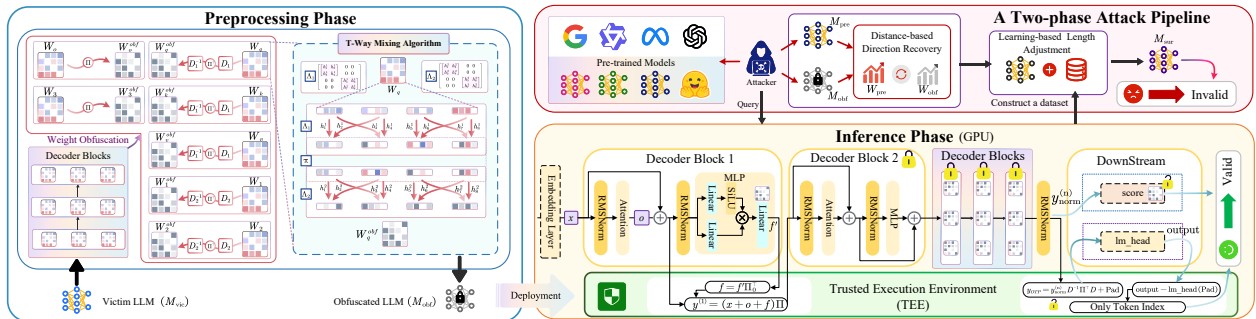

*Figure 1.* **SLIM Workflow.** The red box represents the current state-of-the-art attack pipeline, which we describe in detail in Section 5.1.

TEE-GPU transfer overhead. We summarize and analyze existing TEE-based model obfuscation schemes and compare them in Table 1.

CoreGuard also uses propagation-style execution to reduce TEE authorization to a single initial step. However, its permutation-based transformation does not sufficiently hide vector directions. SLIM preserves this propagation efficiency, but replaces the permutation with a direction-hiding orthogonal mixing transformation to better resist recent direction-recovery attacks.

## 3. Threat Model

**Adversary's Goal.** The adversary aims to steal the functionality of a deployed proprietary model, either for direct use or resale. Specifically, the adversary seeks to construct a surrogate model $M_{\text{sur}}$ that is functionally equivalent to the victim model $M_{\text{vic}}$.

**Adversary's Capabilities.** We assume that the adversary can fully control the device on which the model is deployed and has access to all models and data outside the TEE. However, the adversary cannot access any code or data residing within the TEE. The adversary may leverage publicly available pre-trained models $M_{\text{pre}}$ and datasets to recover the obfuscated model weights. We further assume that the adversary can obtain a small amount of labeled data by querying $M_{\text{vic}}$ to train $M_{\text{sur}}$ for the attack. This assumption is widely adopted in realistic model extraction scenarios (Hua et al., 2018; Yan et al., 2020; Rakin et al., 2022). Notably, our threat model is consistent with prior work summarized in Table 1. We do not address specific security challenges targeting TEEs, such as side-channel attacks.

## 4. Our Proposed Scheme: SLIM

We design an efficient TEE-based secure inference framework for LLMs, termed SLIM, whose overall workflow is illustrated in Figure 1. The framework consists of two phases: **(1) Preprocessing phase:** given the victim model $M_{\text{vic}}$, we apply a weight transformation using a specially constructed

matrix $\Pi$ (generated as described in Algorithm 1) to obtain an obfuscated model $M_{\text{obf}}$. The obfuscated model $M_{\text{obf}}$ is then deployed on the device to perform inference. **(2) Secure inference phase:** during inference, after the first Decoder Block produces its output, the model obtains authorization from the TEE, and the authorization state is retained throughout the entire inference process (see Section 4.1). Ultimately, SLIM ensures that only inference executions authorized by the TEE can produce valid outputs.

### 4.1. Secure Inference Scheme

SLIM builds on the propagation-style authorization paradigm, where a transformed state is authorized once and then propagated through subsequent layers without repeated TEE recovery. Our key modification is to make this propagation compatible with a direction-hiding orthogonal transformation rather than a permutation-based transformation.

To ensure the propagatability of the authorized data flow, the authorization mechanism should satisfy $\text{Decoder}_{\text{obf}}^{(i)}\big(g(y^{(i-1)})\big) = g\big(y^{(i)}\big)$, where $\text{Decoder}_{\text{obf}}^{(i)}$ denotes the $i$-th obfuscated Decoder Block, $y^{(i)}$ denotes the output of the $i$-th original Decoder Block, and $g$ is an invertible authorization function satisfying $g^{-1}\big(g(y^{(i)})\big) = y^{(i)}$. As implied by the above relation, once authorization is performed a single time, the outputs of all subsequent Decoder Blocks retain the authorization information, and the TEE no longer needs to participate in the computation.

**Our Insight:** Orthogonal matrices $\Pi$ possess the two properties: (1) Invertibility: For any $\Pi$, $\Pi \cdot \Pi^{\top} = I$ holds, where $I$ is the identity matrix. (2) Norm-preserving: $||x||_2 = ||x\Pi||_2$. Taking RMSNorm (Zhang & Sennrich, 2019), a normalization technique commonly used in current LLMs, as an example: $\text{RMSNorm}(x) = \gamma \odot \frac{x}{\text{RMS}(x)}$, where $\odot$ denotes the Hadamard (element-wise) product. Let $\gamma = (a_1, a_2, \ldots, a_d)$ and construct the diagonal matrix $D = \text{diag}(a_1, \ldots, a_d)$. Transforming the original input $x$ to $x' = x\Pi$, we have $\text{RMSNorm}(x\Pi) = \frac{x\Pi}{\text{RMS}(x\Pi)} \cdot D = \text{RMSNorm}(x)D^{-1}\Pi D$. Based on the insight, we design

the inference procedure of SLIM. We first formalize the original Decoder Block as follows, where $\sigma$ denotes the activation function.

---

**Decoder Block Formalization**

$$y = \gamma_1 \odot \frac{x}{\text{RMS}(x)}$$

$$Q = yW_q + b_q, K = yW_k + b_k, V = yW_v + b_v$$

$$o = \text{attn}(Q, K, V)W_o + b_o$$

$$z = \gamma_2 \odot \frac{(x + o)}{\text{RMS}(x + o)}$$

$$f = (\sigma(zW_1 + b_1) \otimes (zW_2 + b_2))W_3 + b_3$$

$$output = x + o + f$$

---

**Our Inference Scheme:** We perform obfuscation using orthogonal matrices $\Pi$. For a Decoder Block, with input and output $\in \mathbb{R}^{l \times d}$, it satisfies a function $f_w(x) : \mathbb{R}^{l \times d} \to \mathbb{R}^{l \times d}$. For a given Decoder Block, the model weights are transformed using an obfuscation transformation matrix $\Pi$ and diagonal matrices $D$:

$$W'_q = D_1^{-1}\Pi^\top D_1 W_q, W'_k = D_1^{-1}\Pi^\top D_1 W_k$$
$$W'_v = D_1^{-1}\Pi^\top D_1 W_v, W'_o = W_o\Pi, b'_o = b_o\Pi$$
$$W'_1 = D_2^{-1}\Pi^\top D_2 W_1, W'_2 = D_2^{-1}\Pi^\top D_2 W_2 \quad (1)$$
$$W'_3 = W_3\Pi, b'_3 = b_3\Pi$$

Here, $D_1$ and $D_2$ are the diagonal matrices induced by the RMSNorm weight $\gamma_1$ and $\gamma_2$ in the current Decoder Block. Since we require $D_i^{-1}$, we replace any zero entries in $\gamma_i$ with small non-zero constants to ensure invertibility without affecting inference up to numerical precision. Only the transformed input $x' = x\Pi$ can yield the transformed output $output \cdot \Pi$, ensuring that $\text{Decoder}_{\text{obf}}^{(i)}\big(g(y^{(i-1)})\big) = g(y^{(i)})$ holds and thereby enabling the propagation of authorization. We formalize the resulting obfuscated Decoder Block as follows:

---

**SLIM Decoder Block Formalization**

$$x' = x\Pi$$

$$y' = \gamma_1 \odot \frac{x\Pi}{\text{RMS}(x\Pi)} = \gamma_1 \odot \frac{x\Pi}{\text{RMS}(x)} = yD_1^{-1}\Pi D_1$$

$$Q' = yD_1^{-1}\Pi D_1 D_1^{-1}\Pi^\top D_1 W_q + b_q = Q$$

$$K' = yD_1^{-1}\Pi D_1 D_1^{-1}\Pi^\top D_1 W_k + b_k = K$$

$$V' = yD_1^{-1}\Pi D_1 D_1^{-1}\Pi^\top D_1 W_v + b_v = V$$

$$o' = \text{attn}(Q', K', V')W_o\Pi + b_o\Pi = o\Pi$$

$$z' = \gamma_2 \odot \frac{(x + o)\Pi}{\text{RMS}((x + o)\Pi)} = zD_2^{-1}\Pi D_2$$

$$f' = (\sigma(zD_2^{-1}\Pi D_2 D_2^{-1}\Pi^\top D_2 W_1 + b_1)\otimes$$
$$(zD_2^{-1}\Pi D_2 D_2^{-1}\Pi^\top D_2 W_2 + b_2))W_3\Pi + b_3\Pi = f\Pi$$

$$ouput' = x' + o' + f' = output \cdot \Pi$$

---

*LayerNorm.* For a LayerNorm layer (Ba et al., 2016), we formalize it as:

$$\text{LayerNorm}(x) = \gamma \odot \frac{x - \mu_x \cdot \mathbf{1}_d^\top}{\sqrt{\sigma_x^2 + \epsilon}} + \beta,$$
$$\mu_x = \frac{1}{d}x\mathbf{1}_d, \sigma_x^2 = \frac{1}{d}||x - \mu_x\mathbf{1}_d^\top||_2^2 \quad (2)$$

In the above equation, $x$ is a row vector of size $d$, and $\mathbf{1}_d$ is the all-ones column vector. After applying an orthogonal transformation to the input, $x' = x\Pi$, the matrix $\Pi$ must satisfy the property of preserving $\mathbf{1}_d$, i.e., $\Pi\mathbf{1}_d = \mathbf{1}_d$ and $\mathbf{1}_d^\top\Pi = \mathbf{1}_d^\top$. Then:

$$\mu_{x\Pi} = \frac{1}{d}x\Pi\mathbf{1}_d = \mu_x,$$
$$\sigma_{x\Pi}^2 = \frac{1}{d}||x\Pi - \mu_{x\Pi}\mathbf{1}_d^\top||_2^2 = \frac{1}{d}||x\Pi - \mu_x\mathbf{1}_d^\top\Pi||_2^2 = \sigma_x^2 \quad (3)$$

Therefore, we can obtain a form similar to RMSNorm and handle it in the same way:

$$\text{LayerNorm}(x\Pi) = \gamma \odot \frac{(x - \mu_x \cdot \mathbf{1}_d^\top)\Pi}{\sqrt{\sigma_x^2 + \epsilon}} + \beta \quad (4)$$

To make LayerNorm directly applicable to the SLIM decoder computation, we can transform the bias term $\beta$ accordingly as $\beta' = \beta D^{-1}\Pi D$. Then, $\text{LayerNorm}'(x\Pi) = \text{LayerNorm}(x)D^{-1}\Pi D$, where $\text{LayerNorm}'$ denotes LayerNorm with the transformed bias term.

*Post-RMSNorm.* For the PostNorm architecture, we design an execution pipeline that follows a *branch-first, merge-later* paradigm, as illustrated in Figure 2. Specifically, we duplicate the linear layer preceding the PostNorm block and obfuscate its weights using two different strategies. One branch applies weight transformations based on the specified $\Pi$ and $\gamma$, while the other branch performs transformation using a randomly generated orthogonal matrix $U$, obtained by applying QR decomposition to a random matrix whose entries are sampled from the standard normal distribution. After the computations are completed, the transformed features from both branches are merged to produce the final output, thereby ensuring accurate and consistent propagation of inference results. The introduction of the random orthogonal matrix $U$ does not affect the correctness of the feature transformation. Meanwhile, since $U$ sufficiently mixes the vector directions of the transformed weight matrix, it does not weaken security and further strengthens the obfuscation effect.

**TEE Authorization:** To prevent the exposure of the obfuscation transformation matrices during inference, we apply corresponding obfuscation to the input. Specifically, we obfuscate the down-projection layer of the MLP in the first

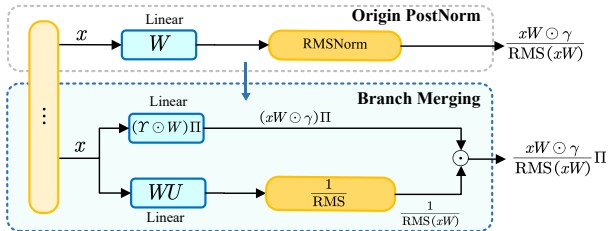

*Figure 2.* The branch-and-merge design of the Post-RMSNorm

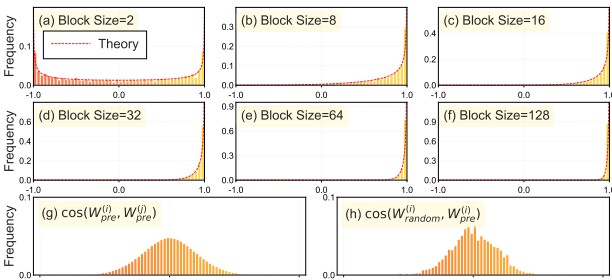

*Figure 3.* Statistical histograms of cosine-similarity distributions. (a)–(f) show the cosine similarities between vectors from $M_{\mathrm{vic}}$ and $M_{\mathrm{pre}}$ on Qwen3-4B after a single $\Lambda$ transformation with different block sizes. (g) shows the cosine-similarity distribution among different vectors within $M_{\mathrm{pre}}$, and (h) shows the cosine-similarity distribution between a random-weight model and $M_{\mathrm{pre}}$.

Decoder Block, i.e., $W_3' = W_3\Pi_0$ and $b_3' = b_3\Pi_0$. The obfuscated layer is then offloaded to the GPU for computation. Subsequently, within the TEE, we perform a corresponding transformation on the output of the first Decoder Block to introduce the transformation matrix used in subsequent obfuscation. This computation is as follows:

$$\text{output}' = (\underbrace{x + o + f\Pi_0\Pi_0^\top})\Pi \tag{5}$$
$$\text{GPU}$$

We prove in Appendix B that an attacker cannot derive $\Pi$ from the intermediate feature information. SLIM applies obfuscation to both linear and nonlinear layers.

**Output Processing:** To prevent the inference outputs from leaking the obfuscation transformation matrices, we apply corresponding obfuscation and recovery operations to the outputs. Different tasks adopt different secure output recovery mechanisms, and whether recovery is performed inside the TEE depends on the type of prediction head.

**For prediction heads with non-shared weights**, such as those used in sequence classification tasks, we directly obfuscate the prediction head by left-multiplying its weights with $D^{-1}\Pi^\top D$, where $D$ represents the diagonalized form of the weights of the final RMSNorm. Performing inference with the obfuscated prediction head yields correct prediction results. **For prediction heads with shared weights**, such as in causal language models, directly applying classification with an obfuscated prediction head would lead to incorrect behavior in the embedding layer. To obtain correct results securely, we introduce a One-Time Pad (OTP) encryption scheme to protect the outputs. Specifically, the output of the backbone network is recovered inside the TEE and combined with random noise before being forwarded to the prediction head for computation. Finally, the TEE removes the effect of the noise and returns only the predicted indices, as illustrated in Figure 1. Note that adding and removing the $\mathrm{Pad}$ incur only lightweight overhead, since generating $\mathrm{Pad}$ and computing $\mathrm{lm\_head}(\mathrm{Pad})$ are independent of the current inference data and can be performed asynchronously inside the TEE while the GPU executes the forward pass.

### 4.2. T-Way Mixing Algorithm

Based on the inference design in Section 4.1, the TEE needs to perform a constant number of $\Pi$-involved matrix–matrix multiplications. When the orthogonal matrix is dense, the computational cost of matrix–matrix multiplication is $O(ld^2)$, which still incurs substantial overhead for the TEE. To minimize TEE computational overhead, we introduce the orthogonal matrix, Householder matrix $H = I - 2vv^\top$, where $v$ is a unit vector. Using a Householder matrix allows matrix-matrix multiplication to be converted into matrix-vector multiplication, i.e., $XH = X - 2(Xv)v^\top$, reducing the computational complexity within the TEE to $O(ld)$. The generation of Householder matrices is completed during the preprocessing phase and therefore does not affect the inference phase.

However, in high-dimensional spaces, the numerical values of a Householder matrix approximate an identity matrix $I$, which cannot effectively obfuscate the weight matrix. Therefore, we propose using a block-diagonal Householder matrix $\Lambda$ for weight obfuscation. Let the block size be $k$, and $\Lambda = \mathrm{diag}(H_1, H_2, \ldots, H_{d/k})$. We observe that the block size significantly affects the obfuscation effectiveness.

**Obfuscation Effectiveness:** In weight matrix obfuscation using left- or right-multiplication, the transformed row (or column) vectors are rotated and scaled versions of the originals. Scaling can be easily inverted using magnitude information from pretrained weights, whereas sufficiently strong rotations destroy directional information, thereby preventing weight recovery that relies on vector directions. Thus, obfuscation effectiveness relies on obfuscating the directional information of the vectors. For left-multiplication, our goal is to achieve $\cos((\Pi^\top W_{\mathrm{vic}})^{(i)}, W_{\mathrm{pre}}^{(i)}) \approx \cos(W_{\mathrm{random}}^{(i)}, W_{\mathrm{pre}}^{(i)})$, where $i$ denotes the row index, and $W_{\mathrm{vic}}$, $W_{\mathrm{pre}}$, and $W_{\mathrm{random}}$ correspond to the weight matrices of the victim model, the pre-trained model, and a random matrix. Through deriva-

tion (Appendix C.1), the probability density function of $\cos((\Lambda^{\top}W_{\text{vic}})^{(i)}, W_{\text{pre}}^{(i)})$ is approximately:

$$f(x) = \frac{\Gamma\left(\frac{k}{2}\right)}{2^{k/2-1}\sqrt{\pi}\,\Gamma\left(\frac{k-1}{2}\right)} \frac{(1+x)^{(k-3)/2}}{(1-x)^{1/2}}, x \in [-1,1] \quad (6)$$

where $\Gamma$ is the Gamma function. We conducted an empirical analysis using Qwen3-4B, and the results are shown in Figure 3. The results align with the theoretical values predicted by the probability density function. Notably, under a single covering operation, there exists no positive integer block size $k$ that meets the aforementioned obfuscation-effectiveness criterion. Furthermore, as the block size $k$ increases, the distance between the original vectors and the obfuscated vectors becomes smaller, implying that overly large blocks diminish the obfuscation effect.

---

**Algorithm 1** T-Way Mixing $\Pi$ Generation Algorithm
___

**Input:** Dimension $d$, Block size $k$, Probability $p$
**Output:** Transformation matrix $\Pi,\{\mathbf{v}^{(j)}\},\{\mathbf{p}^{(j)}\}$
$t \leftarrow \left\lceil \log_k d + \log_k \left(\ln \frac{d^2}{1-p}\right)\right\rceil, \Pi \leftarrow I_d$
**for** $j = 1 \rightarrow t$ **do**
  Initialize $\mathbf{v}^{(j)} \leftarrow [\ ]$
  **for** $i = 1 \rightarrow d/k$ **do**
    Generate $\mathbf{u} \in \mathbb{R}^k$ with entries from $\mathcal{N}(0,1)$
    **if** has LayerNorm **then**
      $\mathbf{u}[k] = -\sum_{q=1}^{k-1}\mathbf{u}[q]$ // Ensure $\Pi\mathbf{1}_d = \mathbf{1}_d$
    **end if**
    $\mathbf{v}_i^{(j)} \leftarrow \mathbf{u}/\|\mathbf{u}\|_2$ and append $\mathbf{v}_i^{(j)}$ to $\mathbf{v}^{(j)}$
  **end for**
  $\mathbf{p}_j \leftarrow \text{randperm}(d)$
  $\Lambda_j \leftarrow \text{diag}(I_{d/k} - 2\mathbf{v}_1^{(j)}\mathbf{v}_1^{(j)\top}, \ldots, I_{d/k} - 2\mathbf{v}_{d/k}^{(j)}\mathbf{v}_{d/k}^{(j)\top})$
  $\Pi \leftarrow \Pi\Lambda_j[:, \mathbf{p}_j]$
**end for**
**Return** $\Pi,\{\mathbf{v}^{(j)}\},\{\mathbf{p}^{(j)}\}$
___

To thoroughly obscure the information of all private vectors, we believe that combining all the vectors (The Obfuscating vector is a linear combination of all vectors) is an effective approach, and the theoretical proof is provided in the Appendix C.2. However, using diagonal block matrices for transformation can only mix the vectors within a single block operation. Therefore, we propose the T-Way Mixing Algorithm, which performs a new mixing after each round of shuffling through a permutation matrix. It can be shown that after at least $\lceil \log_k d \rceil$ rounds, the new vectors can all contain all the original vectors. To simplify arrangements in a scrambled order, we demonstrated that random permutation achieves the same effect with probability $p$ in just $\left\lceil \log_k d + \log_k \left(\ln \frac{d^2}{1-p}\right)\right\rceil$ rounds (optimal rounds, see Appendix D for the derivation). We perform the computation inside the TEE using the factorized form of $\Pi$, and Algorithm 1 presents the construction of $\Pi$ and its factorization.

# 5. Analysis and Experiments

## 5.1. Security Analysis

**Models and Datasets.** To evaluate SLIM, we selected four LLMs that cover the entire architecture of Section 4.1: GPT2-Small (124M parameters) (Radford et al., 2019), Gemma3-1B (Team et al., 2025a), LLaMA3.2-3B (Dubey et al., 2024), and Qwen3-4B (Yang et al., 2025). The datasets cover multiple domains. For the GPT2-Small model, we selected four datasets from the GLUE benchmark (Wang et al., 2018): MNLI (Williams et al., 2018), QQP (Shankar et al., 2017), SST-2 (Socher et al., 2013), and QNLI (Rajpurkar et al., 2016). For Gemma3-1B, LLaMA3.2-3B, and Qwen3-4B, we selected four more complex datasets: WIC (Word-in-Context) (Pilehvar & Camacho-Collados, 2019), GoEmotions (Demszky et al., 2020), FinQA (Chen et al., 2021), and PubMedQA (Jin et al., 2019). For GoEmotions, we use the F1-score as the metric. For FinQA, we use execution accuracy. For all other datasets, we use accuracy.

**Defense Effectiveness.** The current state-of-the-art attack (Wang et al., 2025) exploits the distributional similarity between a public pre-trained model and an obfuscated model to recover weights at low cost, posing a core threat to existing TEE-based obfuscation schemes. The attack pipeline is shown in Figure 1. It first leverages the distance similarity between $M_{\text{pre}}$ and $M_{\text{obf}}$ to recover an initial model $M_{\text{init}}$, and then fine-tunes $M_{\text{init}}$ using data obtained by querying $M_{\text{obf}}$ to construct a surrogate model $M_{\text{sur}}$. Notably, this attack breaks protection directly via statistical analysis of model features, without observing or exploiting inference input–output pairs. We compare the defense results against attacks with previous methods. Building on prior research, we adopt four baseline methods for comparison: (1) No-Shield: The attacker can directly steal the model architecture and weights, i.e., $M_{\text{sur}} = M_{\text{vic}}$. (2) Shield-Whole: The attacker can only obtain the model architecture and uses 1% of the data collected from $M_{\text{vic}}$ (as in prior work) to fine-tune a public pre-trained model $M_{\text{pre}}$, resulting in $M_{\text{sur}}$. (3) Obfuscation Scheme: The attacker uses $M_{\text{obf}}$ as $M_{\text{init}}$, trains with 1% of constructed data, and obtains $M_{\text{sur}}$. (4) Random: Represents the scenario where the attacker can only use a randomly initialized model as $M_{\text{init}}$.

For SLIM, we set the Householder block size to 4 and the round of transformations to the optimal round calculated by the formula in the Algorithm 1 ($p = 0.99$). For SOTER, following the original paper, we randomly select 20% of the weights for obfuscation. For TSQP, we use the loss function provided in the original TSQP paper to train the model, and perform a grid search for the two regularization coefficients $\alpha$ and $\beta$ to ensure optimal model training. We use the AdamW optimizer and unified training hyperparameters on a server with four NVIDIA RTX 5090 GPUs.

*Table 2.* **Evaluation of defense performance under state-of-the-art statistical analysis attack.** We report the model accuracy of each obfuscation scheme $M_{\text{obf}}$ before and after being subjected to the state-of-the-art statistical analysis attack.

| Scheme | GPT2-Small | | | | Gemma3-1B | | | | LLaMA3.2-3B | | | | Qwen3-4B | | | |
|---|---|---|---|---|---|---|---|---|---|---|---|---|---|---|---|---|
| | MNLI | QQP | SST-2 | QNLI | WIC | GoEmotions | FinQA | PubMedQA | WIC | GoEmotions | FinQA | PubMedQA | WIC | GoEmotions | FinQA | PubMedQA |
| **SOTER** | 34.58 | 60.35 | 54.70 | 51.12 | 63.91 | 53.22 | 39.90 | 77.33 | 54.22 | 8.44 | 0 | 0 | 48.59 | 8.29 | 0 | 0 |
| ☠ | 81.27 | 88.37 | 89.70 | 86.18 | 67.19 | 52.29 | 36.86 | 82.67 | 74.06 | 55.29 | 53.24 | 79.33 | 73.13 | 53.54 | 67.24 | 88.00 |
| **ShadowNet** | 31.74 | 62.16 | 49.08 | 50.53 | 50.31 | 7.83 | 0 | 0 | 48.91 | 7.52 | 0 | 0 | 49.53 | 7.76 | 0 | 0 |
| ☠ | 81.25 | 88.39 | 90.60 | 86.00 | 67.34 | 52.19 | 36.57 | 82.67 | 73.91 | 55.19 | 53.24 | 78.67 | 73.13 | 53.45 | 67.05 | 88.00 |
| **TranslinkGuard** | 32.84 | 63.19 | 50.92 | 49.47 | 51.88 | 8.78 | 0 | 0 | 49.69 | 6.10 | 0 | 0 | 52.03 | 8.72 | 0 | 0 |
| ☠ | 81.25 | 88.36 | 89.79 | 86.15 | 67.19 | 52.42 | 36.86 | 82.00 | 74.06 | 55.28 | 53.14 | 78.67 | 73.13 | 53.56 | 67.33 | 88.00 |
| **CoreGuard** | 31.83 | 63.19 | 50.92 | 49.47 | 49.22 | 8.38 | 0 | 0 | 50.00 | 4.84 | 0 | 0 | 50.00 | 8.92 | 0 | 0 |
| ☠ | 80.94 | 88.25 | 91.06 | 86.71 | 65.63 | 51.39 | 42.95 | 76.67 | 74.38 | 54.82 | 60.10 | 85.33 | 73.28 | 54.96 | 64.86 | 84.67 |
| **TSQP** | 32.40 | 41.87 | 51.95 | 49.36 | 52.81 | 37.70 | 3.05 | 32.00 | 50.31 | 7.90 | 0 | 0 | 49.84 | 7.55 | 0 | 0 |
| ☠ | 79.75 | 86.92 | 90.02 | 85.74 | 68.59 | 52.56 | 45.05 | 62.67 | 74.22 | 53.36 | 55.62 | 84.00 | 70.94 | 54.13 | 69.52 | 90.00 |
| **SLIM (Ours)** | 33.03 | 36.84 | 53.21 | 49.93 | 49.38 | 7.07 | 0 | 0 | 49.84 | 5.38 | 0 | 0 | 51.09 | 11.00 | 0 | 0 |
| ☠ | **40.65** | **63.19** | **49.08** | **52.51** | **51.09** | **11.18** | **0** | **0** | **47.97** | **7.34** | **0** | **0** | **50.16** | **6.92** | **0** | **0** |
| **No-Shield** | 82.03 | 89.46 | 92.20 | 87.90 | 67.19 | 54.90 | 47.52 | 80.00 | 73.91 | 56.95 | 60.10 | 87.33 | 73.13 | 56.84 | 68.19 | 89.33 |
| **Shield-Whole** | 66.27 | 77.08 | 75.00 | 63.38 | 48.91 | 18.03 | 0.29 | 47.33 | 50.31 | 18.18 | 14.67 | 61.33 | 54.69 | 20.30 | 7.24 | 60.67 |
| **Random** | 35.83 | 68.37 | 58.48 | 54.43 | 50.47 | 9.09 | 0 | 0 | 50.46 | 2.58 | 0 | 0 | 51.41 | 7.61 | 0 | 0 |

[1] For each method, the first row reports the accuracy of the obfuscated model under the corresponding scheme, whereas the second row (marked with ☠) reports the model accuracy after the attack.
[2] Bold numbers represent the lowest model accuracy after the attack for the corresponding obfuscation scheme on a given dataset (indicating the optimal defense effect).

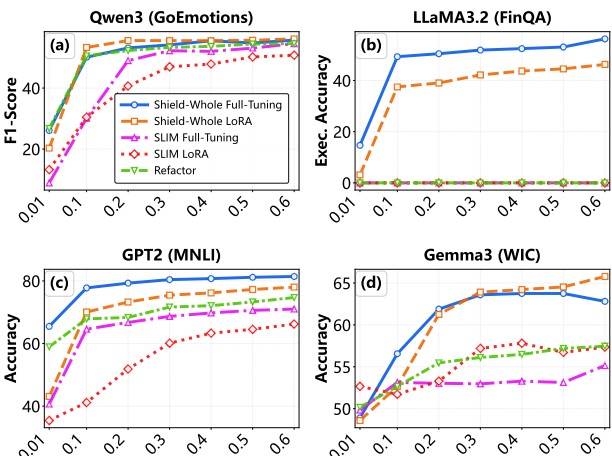

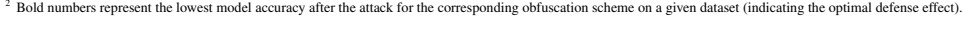

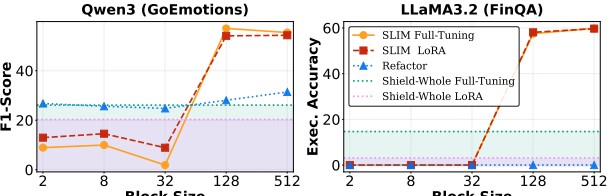

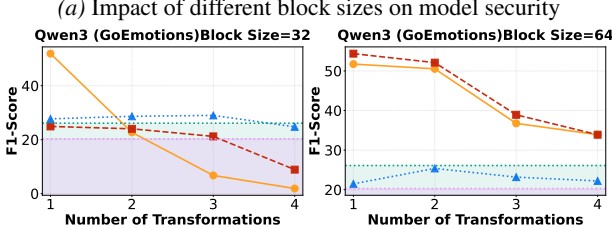

*Figure 4.* Attack defense performance under different budgets.

*(a)* Impact of different block sizes on model security

*(b)* Model security under different numbers of transformations
*Figure 5.* Comparison of factors influencing model security

Table 2 shows the accuracy of the models recovered by the attacker after implementing directional statistical analysis attacks on each obfuscation scheme. It can be observed that previous schemes can be quickly restored to a level close to No-Shield through attacks, while the accuracy of models recovered after attacking SLIM is generally lower than that of Shield-Whole, approximating the defense result of Random, which is equivalent to the attacker fine-tuning a randomly initialized LLM.

**Resilience to Various Attacks.** Assuming the attacker has stronger data collection capabilities, they can query $M_{\text{obf}}$ to obtain a large amount of labeled data, up to 60% of the original training data, and perform fine-tuning attacks using these data. We considered both full-parameter fine-tuning (Full-Tuning) and LoRA-based fine-tuning (Hu et al., 2022), and Π reconstruction attack, where the attacker trains a module that functions the same as the TEE. In the reconstruction attack, the attacker freezes all weights except for the last

linear layer of the first Decoder Block and adds a linear layer after the first Decoder Block. This linear layer attempts to fit a Π to transform the output of the first Decoder Block. The experimental results are shown in Figure 4. It can be observed that even as the amount of data constructed by the attacker increases, the accuracy of SLIM under both Full-Tuning and LoRA is significantly lower than that of Shield-Whole. For the Π reconstruction attack, the effect is similar to Shield-Whole in Figure 4(a), but much lower than Shield-Whole in Figure 4(b) due to the higher complexity of the FinQA dataset. Overall, the defense effectiveness of SLIM remains comparable to Shield-Whole even when the attacker has greater data collection capabilities for fine-tuning attacks and Π reconstruction attacks.

**Impact of Different Block Sizes on Model Security.** We validated the final obfuscation accuracy under the optimal number of transformations for different block sizes of $\Lambda$, as

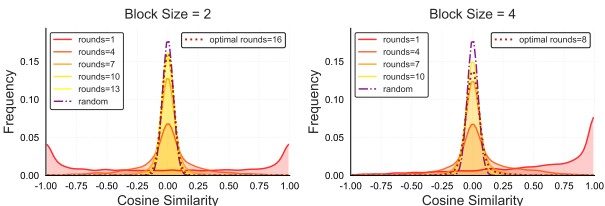

*Figure 6.* Obfuscation effectiveness analysis based on the Qwen3

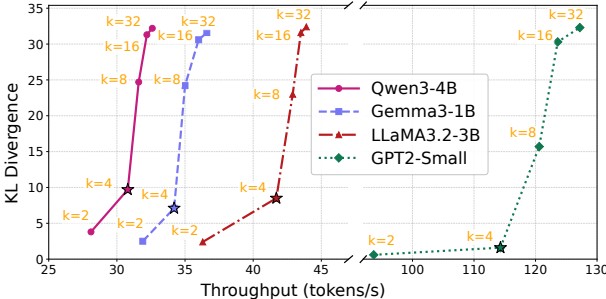

*Figure 7.* Throughput–Obfuscation Effectiveness Trade-off under Different Block Sizes

shown in Figure 5a. We observe that when the block size is 2, 8, or 32, the defense effect reaches that of Shield-Whole. We observed that when the block size is too large, the effect of the attack increases significantly. Therefore, consistent with our theoretical analysis in Section 4.2, SLIM cannot choose a large block size; otherwise, it cannot guarantee obfuscation effectiveness.

**Impact of Different Numbers of Transformations on Model Security.** We validated model security for two block sizes (block size=32, 64), as shown in Figure 5b. We observe that as the number of transformations increases, model security significantly improves.

**Obfuscation Effectiveness**. As shown in Figure 6, with block size 2 or 4 and $p = 0.99$, applying the transformation for the optimal rounds effectively conceals vector direction, making $\cos(w_{\text{pre}}, w_{\text{obf}}) \approx \cos(w_{\text{pre}}, w_{\text{random}})$. We further quantify this distributional closeness using $KL(\cos(w_{\text{pre}}, w_{\text{obf}}), \cos(w_{\text{pre}}, w_{\text{random}}))$, where a smaller value indicates that the obfuscated weights are closer to random weights in directional distribution. For larger block sizes in Appendix C.3, the distribution of $\cos(w_{\text{pre}}, w_{\text{obf}})$ shifts to the right. For the Post-Norm branch, the QR-generated orthogonal matrix $U$ also effectively disrupts weight directions, yielding a cosine distribution close to that of random weights, with a KL divergence below 0.1.

To further study the security–efficiency trade-off, we report a Pareto analysis over different block sizes in Figure 7. The y-axis uses the above KL divergence, and the x-axis is the throughput measured with 128-token inputs, where each point corresponds to a block size $k$. The results show that larger $k$ improves throughput but increases the KL divergence. By the elbow principle, $k = 4$ offers a favorable balance, and we use it as the default block size.

### 5.2. Inference Efficiency Analysis

To evaluate the efficiency of SLIM in actual inference, we conducted experiments on a server equipped with Intel SGX (Costan & Devadas, 2016), featuring two 4th Gen Intel Xeon Scalable Processors 4410Y and one NVIDIA A100 80G PCIe GPU (Our scheme is applicable to any TEE). For implementation, we used PyTorch and Transformers to build the model on the GPU, and manually implement the

operators in SGX.

A block size of 4 is selected for SLIM, aligning with the preceding analysis. GroupCover employs a block size of 2 to minimize computational overhead. We omit the OTP computation in ArrowCloak and GroupCover. This choice favors these baselines, so our reported gains are conservative with respect to end-to-end overhead. For both schemes, non-linear layers and recovery operations are offloaded to SGX, consistent with their original specifications. We discuss the challenges of implementing a cost-effective OTP for Group-Cover and ArrowCloak in Appendix E. Notably, our scheme SLIM fully implements OTP. We measured the inference efficiency of four causal language models. The TTFT for each model is divided into three parts: TEE computation time, TEE-GPU data transfer time, and GPU computation time. The results are shown in Figure 8.

SLIM delivers the best performance across four models and varying input lengths. Relative to SLIM, ArrowCloak incurs a TTFT overhead of $4.49\times\sim13.80\times$ and achieves only $0.24\times\sim0.54\times$ of SLIM's throughput. GroupCover incurs a TTFT overhead of $5.68\times\sim21.31\times$ and reduces throughput to $0.17\times\sim0.55\times$ of SLIM. On average, ArrowCloak and GroupCover incur $222.88\times$ and $203.03\times$ higher TEE-GPU transfer overhead, as well as $5.09\times$ and $22.51\times$ higher TEE computation overhead than SLIM, respectively. Moreover, the performance gap between SLIM and pure GPU inference is largely insensitive to input length and is driven primarily by model size. As the model size grows, SLIM increasingly approaches pure GPU performance. For example, on GPT2-Small, our TTFT and throughput are, on average, $3.66\times$ and $0.55\times$ those of the Pure-GPU baseline, respectively. On Qwen3-4B, our TTFT and throughput are, on average, $1.38\times$ and $0.82\times$ those of the Pure-GPU baseline, respectively, approaching pure-GPU-level performance.

### 5.3. Model Accuracy Analysis

To evaluate the impact on model accuracy, we use **Top-1 Token Accuracy**, defined as the match rate between the Top-1 token produced by the victim model and that produced

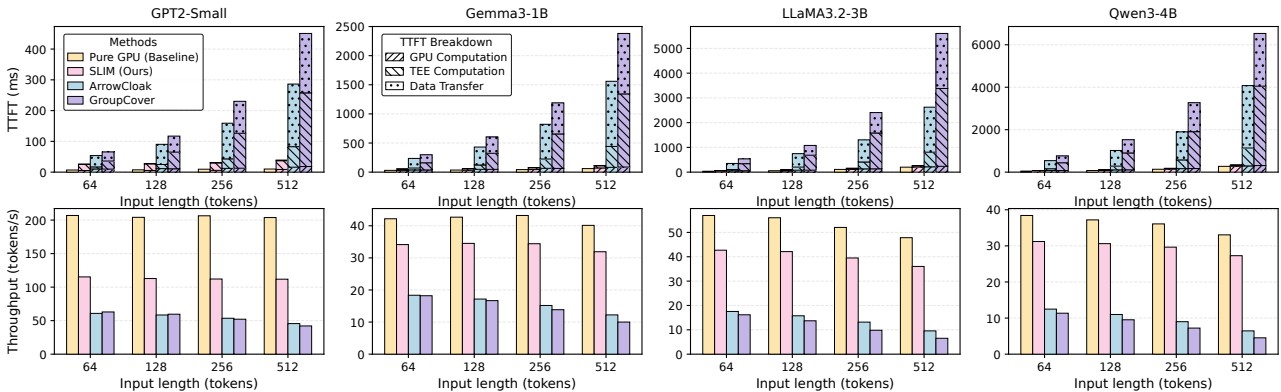

*Figure 8.* The performance of SLIM against three baselines (Pure GPU, ArrowCloak, and GroupCover) in terms of TTFT and end-to-end throughput. Pure GPU reports the inference performance of the victim model $M_{\mathrm{vic}}$ executed on the GPU, whereas the other schemes report secure inference performance for the obfuscated model $M_{\mathrm{obf}}$ using the TEE+GPU pipeline.

*Table 3.* Comparison of Top-1 Token Accuracy

| Model | ArrowCloak | | GroupCover | | SLIM (Ours) | |
|---|---|---|---|---|---|---|
| | FP32 | FP16 | FP32 | FP16 | FP32 | FP16 |
| GPT2 | 98.3% | 1.5% | 99.8% | 71.9% | 100% | 96.7% |
| Gemma3 | 93.5% | 0% | 99.1% | 0% | 100% | 95.2% |
| LLaMA3.2 | 98.4% | 0.6% | 99.9% | 0% | 100% | 99.3% |
| Qwen3 | 91.6% | 0% | 99.4% | 0% | 100% | 99.5% |

under secure inference. Table 3 reports the results under both FP32 and FP16 precision. In FP32, SLIM achieves 100% Top-1 Token Accuracy across all four causal language models, outperforming ArrowCloak and GroupCover. In FP16, ArrowCloak degrades sharply, while GroupCover preserves moderate accuracy only on GPT2 (71.9%), likely because GPT2 is smaller and less sensitive to accumulated numerical errors. However, GroupCover drops to 0% on the other three models. By contrast, SLIM consistently maintains high FP16 accuracy, ranging from 95.2% to 99.5%, showing stronger robustness under low-precision inference.

## 6. Limitations

**Non-cryptographic security guarantee.** Our security analysis is based on the underdetermination of the recovery equations and empirical resistance to existing attacks. Therefore, SLIM does not provide cryptographic-level security guarantees or reductions to standard hardness assumptions.

**Sensitivity to low-precision inference.** SLIM introduces additional transformation and recovery operations. Although these operations preserve Top-1 Token Accuracy in FP32, they may amplify numerical errors in FP16/BF16, leading to accuracy degradation under low-precision inference.

**Lack of quantization support.** The current design does not

support quantized inference. Directly applying INT8/INT4 quantization may break the algebraic properties required by SLIM due to scaling, clipping, and quantization errors.

## 7. Conclusion

In this paper, we propose a novel TEE-based inference framework, SLIM, which provides the highest level of obfuscation effectiveness for LLMs, along with the highest inference efficiency and accuracy. We comprehensively evaluate SLIM against other advanced obfuscation schemes and validate its effectiveness.

## Impact Statement

This work aims to improve the confidentiality and deployment security of large language models on untrusted devices. By mitigating efficient model extraction while preserving inference efficiency, SLIM can help protect legitimate model assets and support practical LLM deployment across edge devices and third-party platforms. However, stronger model-protection mechanisms may also reduce external visibility into deployed models, potentially complicating independent auditing, accountability, and red-teaming. SLIM also does not address all risks, such as black-box extraction, model misuse, or TEE-specific attacks including side channels. Therefore, it should be deployed together with complementary safeguards such as access control, monitoring, auditing interfaces, and TEE hardening.

## Acknowledgment

This work was supported by the Fundamental and Interdisciplinary Disciplines Breakthrough Plan of the Ministry of Education of China (No. JYB2025XDXM118), the National Natural Science Foundation of China (Grant Nos. 62472432, 62306325).

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

# A. Comparison with Existing Works

*Table 4.* Comparison of various weight obfuscation methods.

| Method | Weight Obfuscation | GPU Computing | TEE Restore | Symbol Explanation |
|---|---|---|---|---|
| SOTER | Obfuscate $\theta = 20\%$ layers $W' = g(W) = \mu W \pi$ | $y' = \mu x W \pi = \mu y \pi$ | $y = g^{-1}(y') = y' \pi^\top / \mu$ | $\mu$ is the scaling factor, $\pi$ is permutation matrix |
| ShadowNet | $W' = g(W) = W D \pi$ | $y' = x W D \pi = y D \pi$ | $y = g^{-1}(y') = y' \pi^\top D^{-1}$ | $D$ is a diagonal matrix, $\pi$ is permutation matrix |
| GroupCover | $W = \{w_1, w_2, \ldots, w_d\}$ $W' = g(W) = \{\{w^{(1)}, \ldots, w^{(k)}\}A_1, \ldots, \{w^{(d-k+1)}, \ldots, w^{(d)}\}A_{d/k}\}\pi$ | $y' = x W A \pi = y A \pi$ | $y = g^{-1}(y') = y' \pi^\top A^{-1}$ | GroupCover clusters the original weight column vectors into $d/k$ groups and linearly combines the vectors within the same group, $A_i$ is the coefficient matrix |
| TransLinkGuard | $W' = g(W) = W \pi$ | $y' = x W \pi = y \pi$ | $y = g^{-1}(y') = y' \pi^\top$ | $\pi$ is permutation matrix |
| CoreGuard | $W' = g(W) = W \pi$ | $y' = x W \pi = y \pi$ | $y = g^{-1}(y') = y' \pi^\top$ | $\pi$ is permutation matrix shared by all layers |
| TSQP | $W' = g(W_{dissim}) = \mu W_{dissim} \pi$ | $y' = \mu x W_{dissim} = \mu y$ | $y = g^{-1}(y') = y'/\mu$ | $\mu$ is the scaling factor, $W_{dissim}$ is a learned weight to optimize dissimilarity |
| ArrowCloak | $W' = g(W) = (W D_1 + \mathbf{v} \cdot \mathbf{1}_d D_2)\pi$ | $y' = x(W D_1 + \mathbf{v} \cdot \mathbf{1}_d D_2)\pi$ | $y = (y\pi^\top - x\mathbf{v} \cdot \mathbf{1}_d D_2)D_1^{-1}$ | $D_1$ and $D_2$ are diagonal matrices, $\mathbf{v} = \sum k_j w_j$ |
| SLIM | $W' = g(W) = W \Pi$ | $y' = x W \Pi = y \Pi$ | $y = g^{-1}(y') = y' \Pi^\top$ | $\Pi = \prod^{\log_k d} \Lambda_j \pi_j$, $\Lambda_j$ is a block diagonal Householder matrix with block size $k$ |

Let $g(\cdot)$ denote the obfuscation algorithm. Using a linear layer as an example, the input is $x$, the output is $y$, and the original weight matrix is $W$. We summarize the corresponding computations of each method in Table 4.

In addition, Table 5 summarizes whether existing approaches support mainstream LLM architectures, including GLM-4.7 (Team et al., 2025b), Gemma3, LLaMA4, Qwen3, OLMo2 (OLMo et al., 2025), PaLM (Anil et al., 2023), and Qwen-Next (Qiu et al., 2025).

*Table 5.* Comparison of Model Support

| Mainstream Technology | Representative Models | Compatibility | | | | | | | |
|---|---|---|---|---|---|---|---|---|---|
| | | SOTER | ShadowNet | GroupCover | TransLinkGuard | CoreGuard | TSQP | ArrowCloak | SLIM |
| Post-RMSNorm | OLMo2 | ✓ | ✓ | ✓ | ✓ | ✓ | ✓ | ✓ | ✓ |
| QK-Norm | OLMo2, Qwen3 | ✗ | ✗ | ✗ | ✓ | ✓ | ✗ | ✗ | ✓ |
| Pre+Post RMSNorm | Gemma3 | ✓ | ✓ | ✓ | ✓ | ✓ | ✓ | ✓ | ✓ |
| Pre-RMSNorm | Qwen3, GLM-4.7 | ✓ | ✓ | ✓ | ✓ | ✓ | ✓ | ✓ | ✓ |
| MoE | LLaMA 4 | ✓ | ✓ | ✓ | ✓ | ✓ | ✓ | ✓ | ✓ |
| RoPE | Qwen3 | ✓ | ✗ | ✗ | ✓ | ✓ | ✓ | ✗ | ✓ |
| GeGLU | Gemma3 | ✓ | ✗ | ✗ | ✓ | ✓ | ✓ | ✗ | ✓ |
| SwiGLU | Qwen3 | ✓ | ✗ | ✗ | ✓ | ✓ | ✓ | ✗ | ✓ |
| Sliding-Window Attention | Gemma3 | ✓ | ✓ | ✓ | ✓ | ✓ | ✓ | ✓ | ✓ |
| Gated Attention | Qwen-Next | ✗ | ✗ | ✗ | ✓ | ✓ | ✗ | ✗ | ✓ |
| MHA | OLMo2 | ✓ | ✗ | ✗ | ✓ | ✓ | ✓ | ✓ | ✓ |
| MQA | PaLM | ✓ | ✗ | ✗ | ✓ | ✓ | ✓ | ✓ | ✓ |
| GQA | LLaMA 4 | ✓ | ✗ | ✗ | ✓ | ✓ | ✓ | ✓ | ✓ |
| MLA | DeepSeek-V3/R1 | ✓ | ✗ | ✗ | ✓ | ✓ | ✓ | ✓ | ✓ |

- **QK-Norm** (Henry et al., 2020)**:** Since our scheme computes $Q$ and $K$ correctly on the GPU, applying normalization to $Q$ and $K$ preserves the inference result.

- **RoPE** (Su et al., 2021)**:** Rotary positional embeddings are applied to $Q$ and $K$. Because our scheme produces the correct $Q$ and $K$ on the GPU, it is compatible with RoPE.

- **MoE** (Fedus et al., 2022)**:** Our scheme ensures that the router produces correct outputs. Concretely, the router is typically implemented as an FFN layer; without loss of generality, let $\text{Router}(x) = x W_{\text{router}}$. In our scheme, the

router takes $x\Pi$ as input, and we transform the router weights as $W'_{\text{router}} = \Pi^\top W_{\text{router}}$, yielding

$$\text{Router}'(x\Pi) = x\Pi \cdot \Pi^\top W_{\text{router}} = xW_{\text{router}} = \text{Router}(x).$$

- **MLA:** The first group of equations formalizes the original MLA computation, where the normalized input is projected into query-side and key-value-side latent representations, then split into non-RoPE and RoPE components to construct $Q$, $K$, and $V$ for attention. The second group of equations gives the corresponding SLIM-compatible obfuscated execution. Here, $\Pi$ is the propagated orthogonal transformation generated by the T-Way Mixing $\Pi$ Generation Algorithm, while $\Pi_0$ and $\Pi_1$ are auxiliary random orthogonal matrices, which can be obtained by QR decomposition of random matrices. By transforming the related weights and biases accordingly, SLIM cancels the auxiliary transformations before attention, keeps $Q' = Q$, $K' = K$, and $V' = V$, and preserves the propagation property by producing the obfuscated output $o' = o\Pi$.

---

**MLA Formalization**

$$X = \gamma \odot \frac{x}{\text{RMS}(x)}$$

$$Q_a = XW_{Q_a} + b_{Q_a}$$

$$Q_{norm} = \left(\gamma_1 \odot \frac{Q_a}{\text{RMS}(Q_a)}\right) W_{Q_b} + b_{Q_b}$$

$$[Q_{nr}, Q_r] = \text{Split}(Q_{norm})$$

$$Q = \text{Concat}(Q_{nr}, \text{RoPE}(Q_r))$$

$$[K_{nr}, K_r] = \text{Split}(XW_{KV_a} + b_{KV_a}), \quad W_{KV_a} = \text{Concat}(W_{nr}, W_r), \quad b_{KV_a} = \text{Concat}(b_{nr}, b_r)$$

$$[K_{mid}, V] = \text{Split}\left(\text{Reshape}\left(\left(\gamma_2 \odot \frac{K_{nr}}{\text{RMS}(K_{nr})}\right) W_{KV_b} + b_{KV_b}\right)\right)$$

$$K = \text{Concat}(K_{mid}, \text{RoPE}(K_r))$$

$$o = \text{attn}(Q, K, V) W_o + b_o$$

---

**SLIM MLA Formalization**

$$X' = \gamma \odot \frac{x\Pi}{\text{RMS}(x\Pi)} = XD^{-1}\Pi D$$

$$W'_{Q_a} = D^{-1}\Pi^\top DW_{Q_a}\Pi_0, \quad b'_{Q_a} = b_{Q_a}\Pi_0$$

$$Q'_a = X'W'_{Q_a} + b'_{Q_a} = Q_a\Pi_0$$

$$W'_{Q_b} = D_1^{-1}\Pi_0^\top D_1 W_{Q_b}, \quad b'_{Q_b} = b_{Q_b}$$

$$Q'_{norm} = \left(\gamma_1 \odot \frac{Q'_a}{\text{RMS}(Q'_a)}\right) W'_{Q_b} + b'_{Q_b} = Q_{norm}$$

$$[Q'_{nr}, Q'_r] = \text{Split}(Q'_{norm}) = [Q_{nr}, Q_r]$$

$$Q' = \text{Concat}(Q'_{nr}, \text{RoPE}(Q'_r)) = Q$$

$$W'_{KV_a} = \text{Concat}\left(D^{-1}\Pi^\top DW_{nr}\Pi_1, \ D^{-1}\Pi^\top DW_r\right)$$

$$b'_{KV_a} = \text{Concat}(b_{nr}\Pi_1, \ b_r)$$

$$[K'_{nr}, K'_r] = \text{Split}(X'W'_{KV_a} + b'_{KV_a}) = [K_{nr}\Pi_1, K_r]$$

$$W'_{KV_b} = D_2^{-1}\Pi_1^\top D_2 W_{KV_b}, \quad b'_{KV_b} = b_{KV_b}$$

$$[K'_{mid}, V'] = \text{Split}\left(\text{Reshape}\left(\left(\gamma_2 \odot \frac{K'_{nr}}{\text{RMS}(K'_{nr})}\right) W'_{KV_b} + b'_{KV_b}\right)\right) = [K_{mid}, V]$$

$$K' = \text{Concat}(K'_{mid}, \text{RoPE}(K'_r)) = K$$

$$W'_o = W_o\Pi, \quad b'_o = b_o\Pi$$

$$o' = \text{attn}(Q', K', V')W'_o + b'_o = o\Pi$$

---

- **Activation functions:** Our scheme guarantees that activation functions receive the correct inputs; hence, it is compatible with arbitrary activation functions.

Our method is **KV-cache friendly** because, during secure inference, we allow the attacker to obtain the *correct $K$ and $V$*; this does **not** reveal the model's original weights. Concretely, we can cache $K$ and $V$ directly in GPU memory (VRAM) or in normal host memory.

In contrast, methods such as **SOTER, ShadowNet, GroupCover, TSQP, and ArrowCloak** require TEE-assisted recovery to obtain the correct $K$ and $V$. Moreover, to prevent an attacker from inferring $W_k$ and $W_v$ from the inputs together with the correct $K$ and $V$, these methods must encrypt $K$ and $V$ using the **One-Time Pad (OTP)** before forwarding them to the GPU for computation. Under this design, enabling KV-cache has only two options: (1) **Caching $K$ and $V$ in secure memory**, but secure memory is typically much smaller than normal memory, which can increase paging/swapping and thus overhead. (2) **Caching the obfuscated outputs $K_{\mathbf{obf}}$ and $V_{\mathbf{obf}}$ on the GPU**. When KV-cache is required, $K_{\mathrm{obf}}$ and $V_{\mathrm{obf}}$ must be forwarded to the TEE for de-obfuscation before being used. As the number of generated tokens increases, the KV-cache grows accordingly, which substantially amplifies both the GPU–TEE data transfer overhead and the computational overhead of de-obfuscation within the TEE.

## B. Inference Security Analysis

This section proves that an adversary cannot recover the victim model from the obfuscated weights and intermediate activations observable during secure inference.

### B.1. Security of TEE Authorization

For the first Decoder Block (we ignore the residual connection), an adversary may obtain intermediate computation results through a series of queries, including $x^{(i)}$, $o^{(i)}$, and the gated output $g^{(i)}W_3'$. Meanwhile, the adversary can also observe the obfuscated output $\text{output}^{(i)}\Pi$. Let $X$, $O$, $G$, and $Y$ denote the full-rank square matrices formed by stacking $x^{(i)}$, $o^{(i)}$, $g^{(i)}W_3'$, and $\text{output}^{(i)}\Pi$ row-wise, respectively. Then, we obtain the following system of equations:

$$\begin{cases} Y = (X + O + G\Pi_0^\top)\Pi, \\ W_3' = W_3\Pi_0. \end{cases} \tag{7}$$

Consider constructing $\{W_3 Q, Q^\top \Pi_0\}$, where $Q$ is an orthogonal matrix and $Q \neq I$. It can be verified that $W_3 Q Q^\top \Pi_0 = W_3'$. In this case, $\Pi' = (X + O + G\Pi_0^\top Q)^{-1}Y$. That is, both $\{W_3 Q, Q^\top \Pi_0, \Pi'\}$ and $\{W_3, \Pi_0, \Pi\}$ are valid solutions to the above system. Therefore, the adversary cannot uniquely recover $\Pi$.

### B.2. Security of Intermediate Weight Obfuscation

For intermediate layer computations, without loss of generality, let the sequence length be $1$, and the hidden size be $d$. The adversary may observe values of the form $y^{(i)} = x^{(i)}\Pi$ or $y^{(i)} = \Pi x^{(i)}$, where both $x^{(i)}$ and $\Pi$ are unknown. Taking the first case as an example, we can write the following system of equations:

$$\begin{cases} y^{(1)} = x^{(1)}\Pi, \\ y^{(2)} = x^{(2)}\Pi, \\ \qquad \vdots \\ y^{(N)} = x^{(N)}\Pi. \end{cases} \tag{8}$$

The above system contains $N \cdot d$ equations, whereas the number of unknowns is $N \cdot d + K$, where $K$ depends on the specific form of $\Pi$. Hence, the number of unknowns exceeds the number of equations, rendering the system underdetermined.

Moreover, one can construct two distinct sets of solutions that both satisfy the system, such as $x^{(i)'} = x^{(i)}Q$ and $\Pi' = Q^\top\Pi$, where $Q \neq I$. Therefore, the adversary cannot recover $\Pi$ from the intermediate-layer computation results.

### B.3. Security of Output Processing

We analyze the relation

$$X_{\text{OTP}} = X'\Pi^\top + R, \tag{9}$$

where $X', X_{\text{OTP}}, R \in \mathbb{R}^{l \times d}$ and $\Pi \in \mathbb{R}^{d \times d}$. The adversary observes only $(X', X_{\text{OTP}})$, while the pad $R$ is unobserved.

Fix the observed $(X', X_{\text{OTP}})$. Let $\widetilde{\Pi} \in \mathbb{R}^{d \times d}$ be arbitrary and define

$$\widetilde{R} := X_{\text{OTP}} - X'\widetilde{\Pi}^\top \in \mathbb{R}^{l \times d}. \tag{10}$$

Then

$$X'\widetilde{\Pi}^\top + \widetilde{R} = X'\widetilde{\Pi}^\top + (X_{\text{OTP}} - X'\widetilde{\Pi}^\top) = X_{\text{OTP}}, \tag{11}$$

so (9) holds with $(\widetilde{\Pi}, \widetilde{R})$. Since $\widetilde{\Pi}$ can be chosen arbitrarily, there are infinitely many distinct $\widetilde{\Pi}$ consistent with the same observations $(X', X_{\text{OTP}})$. Therefore, $\Pi$ is not identifiable from $(X', X_{\text{OTP}})$, and no algebraic procedure can uniquely reconstruct it.

# C. Obfuscation Effectiveness Analysis

## C.1. The Effect of a Single Obfuscation Step on Vector Directionality

**Symbol Definition** For clarity, the key notations used throughout this analysis are summarized in Table 6.

*Table 6.* Notation for the single transformation analysis.

| Symbol | Description |
|---|---|
| $d$ | Dimension of the orthogonal matrix, where $\log_d n$ is a positive integer. |
| $O(k)$ | The orthogonal group of order $k$, i.e., the set of all $k \times k$ orthogonal matrices. |
| $W$ | $d \times d$ Haar-random orthogonal matrix, $W \sim \mathrm{Haar}(O(d))$, drawn from the uniform distribution over the orthogonal group. |
| $\{w_i\}_{i=1}^d$ | Row vectors of $W$, forming an orthonormal basis: $\langle w_i, w_j \rangle = \delta_{ij}$ for all $i, j$. |
| $k$ | Group size (block dimension), satisfying $k$ divides $d$ evenly. |
| $H$ | $d \times d$ block-diagonal matrix composed of independent random Householder reflections: $H = \mathrm{diag}(H_1, \ldots, H_{d/k})$. |
| $H_i$ | Random $k \times k$ Householder reflection: $H_i = I_k - 2vv^\top$, where $I_k$ is the $k \times k$ identity matrix. |
| $v$ | $k$-dimensional unit vector uniformly distributed on the $(k-1)$-dimensional unit sphere $S^{k-1} = \{x \in \mathbb{R}^k : \|x\| = 1\}$, denoted $v \sim \mathrm{Uniform}(S^{k-1})$. |
| $W'$ | Transformed orthogonal matrix: $W' = HW$. |
| $w_1'$ | First row vector of $W'$. |
| $\alpha$ | Cosine similarity between $w_1'$ and $w_1$: $\alpha = \langle w_1', w_1 \rangle$. Since both are unit vectors, $\alpha$ equals the cosine of the angle between them. |
| $B(a, b)$ | Beta function: $B(a, b) = \frac{\Gamma(a)\Gamma(b)}{\Gamma(a+b)}$ for $a, b > 0$. |
| $\Gamma(z)$ | Gamma function, satisfying $\Gamma(z) = (z-1)!$ for positive integers $k$, and $\Gamma(1/2) = \sqrt{\pi}$. |

**Problem Statement** Given the orthogonal transformation $W \to W' = HW$ induced by mutually independent, Haar-distributed random Householder blocks $\{H_i\}$, we derive the probability density function (PDF) of the cosine similarity $\alpha = \langle w_1', w_1 \rangle$.

**Assumptions** Without loss of generality, we assume:

1. The initial row vectors $\{w_1, \ldots, w_d\}$ form an orthonormal basis of $\mathbb{R}^d$, i.e., $W$ is an orthogonal matrix. This assumption is an approximation of the empirical observation in Figure 3(g).

2. Each random Householder matrix $H_i$ follows the Haar measure on $O(k)$, and the matrices $\{H_i\}$ are mutually independent.

**Lemma C.1** (Action of the Block-Diagonal Transformation). *Let $H = \mathrm{diag}(H_1, H_2, \ldots, H_{d/k})$ and $W' = HW$. Then the first transformed row vector $w_1'$ depends only on the first block $H_1$ and the first $d$ rows of $W$:*

$$w_1' = \sum_{k=1}^k (H_1)_{1i}\, w_i. \tag{12}$$

*Consequently, the cosine similarity reduces to $\alpha = (H_1)_{11}$.*

*Proof.* Since $H$ is block-diagonal, its first $d$ rows have non-zero entries only in the first $d$ columns, corresponding to the block $H_1$. The matrix multiplication $W' = HW$ implies that the first row of $W'$ is the first row of $H$ multiplied by $W$. Therefore, $w_1' = \sum_{j=1}^d H_{1j}w_j = \sum_{i=1}^k (H_1)_{1i}w_i$, where the last equality uses the block structure $H_{1j} = (H_1)_{1j}$ for $1 \leq j \leq k$ and $H_{1j} = 0$ for $j > k$. By the orthonormality of $\{w_i\}$, we have $\alpha = \langle w_1', w_1 \rangle = \sum_{i=1}^k (H_1)_{1i}\langle w_i, w_1 \rangle = (H_1)_{11} \cdot 1 = (H_1)_{11}$. $\quad\square$

**Lemma C.2** (Distribution of a Squared Coordinate on the Sphere). *Let $v = (v_1, \ldots, v_k)^\top \sim \text{Uniform}(S^{k-1})$. Then the squared coordinate $t = v_1^2$ follows a Beta distribution:*

$$t \sim \text{Beta}\left(\frac{1}{2}, \frac{k-1}{2}\right), \tag{13}$$

*with probability density function*

$$f_T(t) = \frac{1}{B\left(\frac{1}{2}, \frac{k-1}{2}\right)} \, t^{-\frac{1}{2}} (1-t)^{\frac{k-3}{2}}, \quad t \in [0, 1]. \tag{14}$$

*Proof.* This is a standard result from directional statistics: for a uniformly distributed point on the $(k-1)$-sphere, the squared projection onto any axis follows $\text{Beta}\left(\frac{1}{2}, \frac{k-1}{2}\right)$. A derivation follows from representing the uniform distribution on the sphere in terms of independent Gaussian variables and using the relationship between Gamma and Beta distributions. $\square$

**Theorem C.3** (PDF of the Cosine Similarity After a Single Householder Block Transformation). *Under the assumptions stated, the cosine similarity $\alpha = \langle w_1', w_1 \rangle$ induced by a single random Householder block of size $d$ admits the following probability density function:*

$$f_\alpha(\alpha) = \frac{2^{\frac{2-k}{2}}}{B\left(\frac{1}{2}, \frac{k-1}{2}\right)} \, (1-\alpha)^{-\frac{1}{2}} (1+\alpha)^{\frac{k-3}{2}}, \quad \alpha \in [-1, 1]. \tag{15}$$

*Equivalently, in terms of the Gamma function,*

$$f_\alpha(\alpha) = \frac{\Gamma\left(\frac{k}{2}\right)}{2^{\frac{k}{2}-1}\sqrt{\pi}\,\Gamma\left(\frac{k-1}{2}\right)} \, (1-\alpha)^{-\frac{1}{2}} (1+\alpha)^{\frac{k-3}{2}}, \quad \alpha \in [-1, 1]. \tag{16}$$

*Proof.* We proceed by establishing a deterministic relationship between $\alpha$ and $v_1$, followed by a change of variables.

From Lemma C.1, $\alpha = (H_1)_{11}$. By definition, $H_1 = I_k - 2vv^\top$, hence its $(1, 1)$ entry is $(H_1)_{11} = 1 - 2v_1^2$. Therefore,

$$\alpha = 1 - 2v_1^2. \tag{17}$$

Let $T = v_1^2$. By Lemma C.2, $T \sim \text{Beta}\left(\frac{1}{2}, \frac{k-1}{2}\right)$ with density $f_T(t)$ given above. Equation (17) defines a one-to-one transformation $g : T \mapsto \alpha = 1 - 2T$, with inverse $T = (1-\alpha)/2$ and Jacobian $\left|\frac{kt}{k\alpha}\right| = \frac{1}{2}$.

Applying the change-of-variables formula $f_\alpha(\alpha) = f_T(t)\left|\frac{dt}{d\alpha}\right|$ at $t = (1-\alpha)/2$ yields

$$\begin{aligned} f_\alpha(\alpha) &= \frac{1}{B\left(\frac{1}{2}, \frac{k-1}{2}\right)} \left(\frac{1-\alpha}{2}\right)^{-\frac{1}{2}} \left(1 - \frac{1-\alpha}{2}\right)^{\frac{k-3}{2}} \cdot \frac{1}{2} \\ &= \frac{1}{2\,B\left(\frac{1}{2}, \frac{k-1}{2}\right)} \left(\frac{1-\alpha}{2}\right)^{-\frac{1}{2}} \left(\frac{1+\alpha}{2}\right)^{\frac{k-3}{2}}. \end{aligned} \tag{18}$$

Simplifying the powers of 2:

$$\begin{aligned} \left(\frac{1-\alpha}{2}\right)^{-\frac{1}{2}} &= 2^{\frac{1}{2}} (1-\alpha)^{-\frac{1}{2}}, \\ \left(\frac{1+\alpha}{2}\right)^{\frac{k-3}{2}} &= 2^{\frac{3-k}{2}} (1+\alpha)^{\frac{k-3}{2}}. \end{aligned} \tag{19}$$

Multiplying the constants gives $2^{\frac{1}{2} + \frac{3-k}{2} - 1} = 2^{\frac{2-k}{2}}$. Substituting back yields the form in (15).

To obtain the Gamma function form (16), we use the identity $B\left(\frac{1}{2}, \frac{k-1}{2}\right) = \frac{\Gamma(1/2)\,\Gamma\left(\frac{k-1}{2}\right)}{\Gamma\left(\frac{k}{2}\right)}$ and $\Gamma(1/2) = \sqrt{\pi}$. Substituting into (15) and simplifying the factor of $2^{\frac{2-k}{2}}$ completes the derivation. $\square$

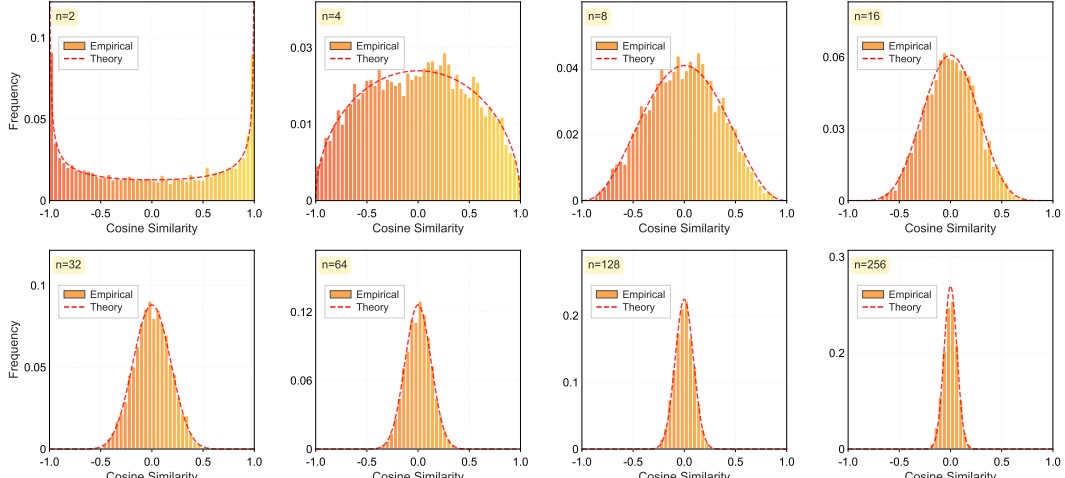

*Figure 9.* The cosine value distribution of the obfuscated vector and the original vector.

**Conclusion**    Theorem C.3 provides the exact distribution of the cosine similarity $\alpha$ after applying a single random Householder block transformation. The PDF depends solely on the block size $k$ and is independent of the overall matrix dimension $d$. The two equivalent expressions, (15) and (16), characterize the obfuscation effect quantitatively.

### C.2. The Effect of Mixing on Vector Directionality

**Theoretical Derivation**    In this section, we prove that when new vectors are generated via linear combinations of multiple vectors, the obfuscation strength of directional information increases as more vectors are mixed. As in the previous section, we introduce several simplifying assumptions without loss of generality:

1. The vectors used for linear combination are mutually orthogonal and have identical norms.

2. The coefficients of the linear combination are randomly generated and are independently and identically distributed (i.i.d.). For example, drawn from a standard normal distribution.

Let $n$ denote the number of vectors involved in the mixing process. It is straightforward to derive the cosine similarity between the mixed vector and an original vector.

$$\alpha = \langle w', w \rangle = \frac{x_1}{\sqrt{x_1^2 + \cdots + x_n^2}} \tag{20}$$

where $x_i$ represents the coefficient of the i-th vector.

Therefore, the original distribution is equivalent to the distribution of the cosine of the angle between a uniformly random unit vector in an $n$-dimensional space and a fixed direction. This distribution admits a well-known closed-form characterization:

$$f(x) = \frac{\Gamma(\frac{n}{2})}{\sqrt{\pi}\Gamma(\frac{n-1}{2})}(1 - x^2)^{\frac{n-3}{2}} \tag{21}$$

**Simulation Results**    We conducted numerous repeated experiments and obtained the frequency distribution histogram. As shown in Figure 9, as the number of mixed vectors increases, their directional information becomes effectively scrambled.

### C.3. Obfuscation Effectiveness

In this section, we experimentally evaluate multiple models to illustrate how the cosine similarity between the obfuscated vectors and the pre-trained vectors—i.e., the obfuscation effectiveness—evolves as the number of obfuscation rounds

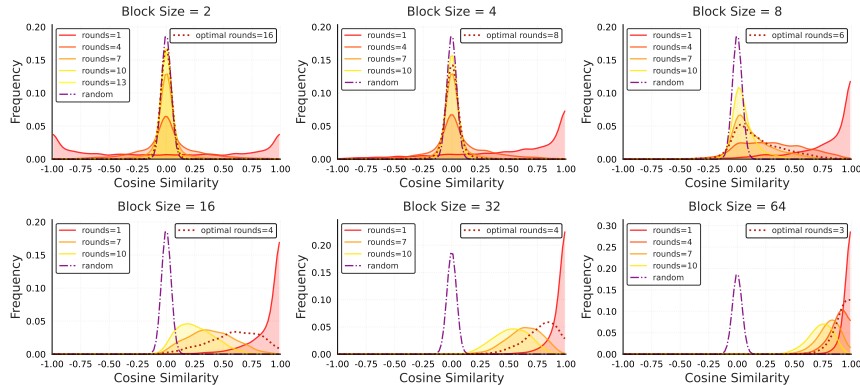

*Figure 10.* Obfuscation effectiveness of Qwen3 under different block sizes across varying numbers of transformation rounds.

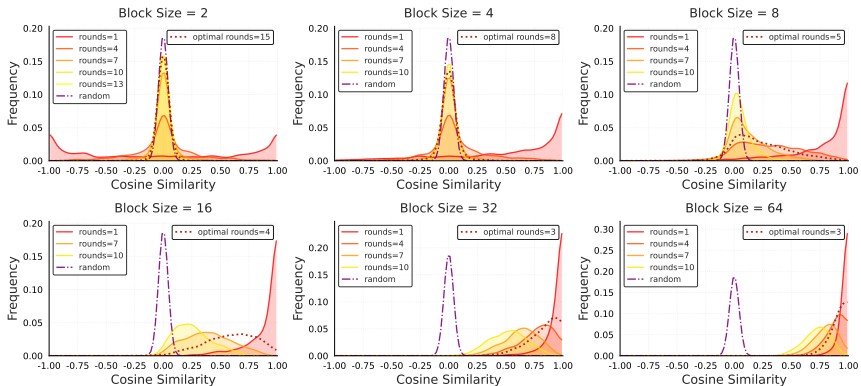

*Figure 11.* Obfuscation effectiveness of LLaMA3.2 under different block sizes across varying numbers of transformation rounds.

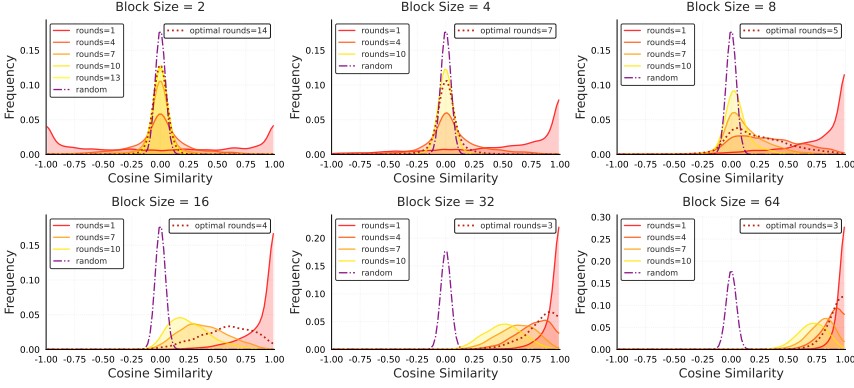

*Figure 12.* Obfuscation effectiveness of Gemma3 under different block sizes across varying numbers of transformation rounds.

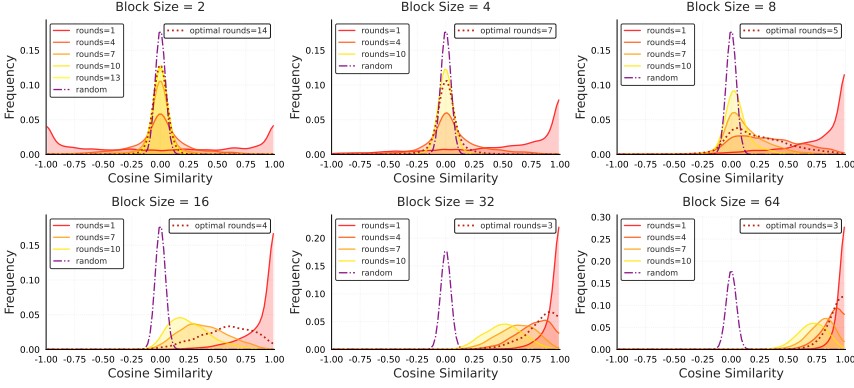

*Figure 13.* Obfuscation effectiveness of GPT2 under different block sizes across varying numbers of transformation rounds.

increases under different block sizes. As shown in Figs. 10–13, our target condition, $\cos(w_{\mathrm{pre}}, w_{\mathrm{obf}}) \approx \cos(w_{\mathrm{pre}}, w_{\mathrm{random}})$, can only be achieved by adopting relatively small block sizes together with a sufficient number of obfuscation rounds.

When the block size is large, the distribution of $\cos(w_{\mathrm{pre}}, w_{\mathrm{obf}})$ exhibits a pronounced right skew. Although this distribution gradually shifts leftward as the number of obfuscation rounds increases, we still recommend using smaller block sizes. This is because the computational overhead inside the TEE depends only on the number of obfuscation rounds and is not directly affected by the block size. The observed right-skew phenomenon arises from the fact that, as the block size increases, random Householder matrices increasingly resemble the identity matrix, which is consistent with the results shown in Figure 3. Meanwhile, we noticed that as the model size increases, $\cos(w_{\mathrm{pre}}, w_{\mathrm{obf}})$ becomes closer to $\cos(w_{\mathrm{pre}}, w_{\mathrm{random}})$ under small block size settings. This is because the vectors before obfuscation do not strictly satisfy pairwise orthogonality (C.1 assumption 1), but as the model size increases, these vectors gradually approach orthogonality.

In practical deployments, we therefore recommend setting the block size to $4$.

# D. Optimal Obfuscation Rounds

We analyze the multi-round grouped mixing process, which can be viewed as uniformly distributing $d$ distinct elements into $d$ bins via repeated random groupings. Our goal is to derive a lower bound on the number of mixing rounds required to guarantee, with high probability, that *every bin contains every element*. The analysis proceeds in two stages: (i) characterizing the decay of the local missing probability for a fixed element–bin pair, and (ii) lifting this local guarantee to a global one via the union bound.

**Notation**  Table 7 summarizes the key symbols used throughout this section.

*Table 7.* Notation for the multi-round obfuscation analysis.

| Symbol | Description |
|---|---|
| $d$ | Number of bins and elements. Sets: $\mathcal{X} = \{x_1, \ldots, x_d\}$ (elements), $\mathcal{B} = \{b_1, \ldots, b_d\}$ (bins). |
| $k$ | Group size (bins per group). We assume $d = kn$ for an integer $n$. |
| $n$ | Number of groups per round: $n = d/k$. |
| $r$ | Number of mixing rounds ($r = 0$ corresponds to the initial state). |
| Mixing rule | Each round partitions $\mathcal{B}$ into $k$ disjoint groups of size $d$ uniformly at random. After intra-group mixing, every bin in a group contains the union of all elements previously present in that group. |
| Initial state | Each element $x \in \mathcal{X}$ resides in exactly one bin $b_x$ initially: $\mathbb{I}(x \in b_x) = 1$, $\mathbb{I}(x \in b) = 0$ for $b \neq b_x$, where $\mathbb{I}(\cdot)$ denotes the indicator function. |
| $\epsilon_r(x, b)$ | Probability that bin $b$ does not contain element $x$ after $r$ rounds. |
| $\epsilon_r$ | Missing probability for a fixed element–bin pair. By symmetry, $\epsilon_r(x, b) = \epsilon_r$ for all $x, b$. |
| $p$ | Target global success probability: $\mathbb{P}(\text{all bins contain all elements}) \geq p$, with $0 < p < 1$. |
| $m_r$ | Expected number of bins containing a fixed element after $r$ rounds. |

**Lemma D.1** (Decay of the local missing probability). *Under the fully random partitioning assumption, the missing probability $\epsilon_r$ for a fixed element–bin pair satisfies the following recurrence relation:*

$$\epsilon_{r+1} = \epsilon_r \cdot \frac{\binom{d-m_r-1}{k-1}}{\binom{d-1}{k-1}}, \tag{22}$$

*where $m_r$ is the expected number of bins containing the fixed element after $r$ rounds. For large $d$ and constant $k$, this simplifies to the approximate recurrence:*

$$\epsilon_{r+1} \leq \epsilon_r \cdot e^{-k^r/d}, \tag{23}$$

*with initial condition $\epsilon_0 = \frac{d-1}{d} \approx 1$ for large $d$. Consequently, after $r$ rounds,*

$$\epsilon_r \leq e^{-k^r/d}. \tag{24}$$

*Proof.* Let $m_r$ denote the expected number of bins that contain a fixed element $x$ after $r$ rounds. Initially, $m_0 = 1$. In each round, any bin containing $x$ is grouped with $k - 1$ other bins, and after intra-group mixing, all $k$ bins in that group will contain $x$. Since groups are formed uniformly at random, each bin containing $x$ can "spread" it to $k - 1$ new bins in expectation. This yields $m_{r+1} \approx k \cdot m_r$, leading to $m_r \approx k^r$.

Now consider a fixed bin $b$ that does not contain $x$ after $r$ rounds. In round $r + 1$, $b$ is grouped uniformly with $k - 1$ other bins from the remaining $d - 1$ bins. The probability that none of these $k - 1$ bins contain $x$ equals the ratio of ways to choose $k - 1$ bins from the $d - m_r - 1$ bins not containing $x$ to the total ways to choose $k - 1$ bins from $d - 1$ bins:

$$\frac{\binom{d-m_r-1}{k-1}}{\binom{d-1}{k-1}}.$$

Thus, $\epsilon_{r+1} = \epsilon_r \cdot \frac{\binom{d-m_r-1}{k-1}}{\binom{d-1}{k-1}}$.

For large $d$, using the approximation $\frac{\binom{d-a}{b}}{\binom{d}{b}} \approx (1 - \frac{a}{d})^b$, we obtain:

$$\frac{\binom{d-m_r-1}{k-1}}{\binom{d-1}{k-1}} \approx \left(1 - \frac{m_r}{d}\right)^{k-1} \leq e^{-m_r(k-1)/d}.$$

Substituting $m_r \approx k^r$, we get the simplified form $\epsilon_{r+1} \leq \epsilon_r \cdot e^{-k^r(k-1)/d}$. Iterating this recurrence from $\epsilon_0 = \frac{d-1}{d} \approx 1$ yields $\epsilon_r \leq e^{-(k^0+k^1+\cdots+k^{r-1})(k-1)/d} = e^{(1-k^r)/d} \approx e^{-k^r/d}$. $\qquad\square$

**Lemma D.2** (Global success probability bound). *After $r$ rounds of mixing, the probability that every bin contains every element is bounded below by*

$$\mathbb{P}(global\ success) \geq 1 - d^2 \epsilon_r, \tag{25}$$

*where $\epsilon_r$ is the local missing probability defined in Lemma D.1.*

*Proof.* For a fixed element $x$, let $A_b = \{x \notin b\}$ be the event that bin $b$ is missing $x$. By the union bound,

$$\mathbb{P}\left(\bigcup_{b \in \mathcal{B}} A_b\right) \leq \sum_{b \in \mathcal{B}} \mathbb{P}(A_b) = \sum_{b \in \mathcal{B}} \epsilon_r = d\epsilon_r.$$

Hence, the probability that $x$ covers all bins is at least $1 - d\epsilon_r$. Equivalently, the probability that $x$ fails to cover all bins is at most $d\epsilon_r$.

Now, let $C_x$ denote the event that element $x$ fails to cover all bins. Applying the union bound again over all elements,

$$\mathbb{P}\left(\bigcup_{x \in \mathcal{X}} C_x\right) \leq \sum_{x \in \mathcal{X}} \mathbb{P}(C_x) \leq \sum_{x \in \mathcal{X}} d\epsilon_r = d \cdot (d\epsilon_r) = d^2 \epsilon_r.$$

Therefore, the probability that no element fails—i.e., all bins contain all elements—is at least $1 - d^2 \epsilon_r$. $\qquad\square$

**Theorem D.3** (Lower bound on required rounds). *To achieve a global success probability of at least $p$ ($0 < p < 1$), the number of mixing rounds $r$ must satisfy*

$$r \geq \log_k \left[ d\big(2 \ln d - \ln(1-p)\big) \right]. \tag{26}$$

*Equivalently,*

$$r \geq \log_k d + \log_k \big(2 \ln d - \ln(1-p)\big).$$

*Proof.* By Lemma D.1 and Lemma D.2, we require $1 - d^2 \epsilon_r \geq 1 - d^2 e^{-k^r/d} \geq p$, which is equivalent to

$$d^2 e^{-k^r/d} \leq 1 - p.$$

Taking natural logarithms on both sides yields

$$\ln(d^2) + \ln\left(e^{-k^r/d}\right) \leq \ln(1-p) \quad \Rightarrow \quad 2 \ln d - \frac{k^r}{d} \leq \ln(1-p).$$

Rearranging terms,

$$\frac{k^r}{d} \geq 2 \ln d - \ln(1-p).$$

Since $\ln(1-p) < 0$ for $0 < p < 1$, the right-hand side is positive for sufficiently large $d$. Multiplying by $d$ and then applying the monotonically increasing function $\log_k(\cdot)$ completes the proof:

$$k^r \geq d\big(2 \ln d - \ln(1-p)\big) \quad \Longrightarrow \quad r \geq \log_k \left[ d\big(2 \ln d - \ln(1-p)\big) \right].$$

$\qquad\square$

**Remark (Exact recurrence and validation)** The exact recurrence for $\epsilon_r$ can be derived from combinatorial considerations. Let $m_r$ be the exact (non-integer) expected number of bins containing a fixed element after $r$ rounds. The recurrence $\epsilon_{r+1} = \epsilon_r \cdot \frac{\binom{d-m_r-1}{k-1}}{\binom{d-1}{k-1}}$ holds because the event that $b$ misses $x$ after $r+1$ rounds requires that $b$ missed $x$ after $r$ rounds and that in round $r+1$, none of the $k-1$ bins grouped with $b$ contain $x$. For large $d$, using the approximation $\binom{d-a}{b} \approx \frac{d^b}{b!}(1-\frac{a}{d})^b$, we have:

$$\frac{\binom{d-m_r-1}{k-1}}{\binom{d-1}{k-1}} \approx \frac{(1-\frac{m_r+1}{d})^{k-1}}{(1-\frac{1}{d})^{k-1}} \approx (1-\frac{m_r}{d})^{k-1} \leq e^{-m_r(k-1)/d}.$$

With $m_r \approx k^r$, we obtain the simplified form used in Lemma D.1.

**Simulation Results** We vary $d$ and $k$ and conduct extensive simulation experiments. Table 8 reports (i) the empirical number of mixing rounds required to achieve a probability of mixing all vectors exceeding 0.99, and (ii) the corresponding theoretical requirement. In each cell, the first entry is the empirical result and the second entry is the theoretical prediction. Overall, the theoretical predictions closely match the empirical results, with only a few cases where the theory slightly overestimates the required rounds. This conservatism arises because proof in Theorem D.3 uses a sufficient condition for achieving $\mathbb{P}(\text{global success})$.

*Table 8.* Results under different parameter combinations.

| d | k=2 | k=8 | k=32 | k=128 | k=256 |
|---|-----|-----|------|-------|-------|
| 512 | (14,14) | (5,5) | (3,3) | (2,2) | (2,2) |
| 1024 | (15,15) | (5,5) | (3,3) | (2,3) | (2,2) |
| 2048 | (16,16) | (6,6) | (3,4) | (3,3) | (2,2) |
| 4096 | (17,17) | (6,6) | (4,4) | (3,3) | (2,3) |

# E. Limitation of OTP-based Encryption

Both GroupCover and ArrowCloak require OTP encryption of the results forwarded from the TEE to the GPU to prevent adversaries from inferring model weights from input–output pairs. It is important to emphasize that the state-of-the-art attack considered in this paper (the security experiments in Section 5.1) uses only the obfuscated model weights, the corresponding pretrained model weights, and a constructed recovery training dataset; it does not leverage per-layer input–output pairs to mount the attack. Therefore, ignoring OTP only makes GroupCover and ArrowCloak appear better in our efficiency evaluation (by omitting the additional overhead introduced by their OTP), and has no impact on the security experiments.

## E.1. OTP Computation Overhead Analysis

For GroupCover, $Q_{\text{obf}} = Q\pi_Q$, $K_{\text{obf}} = K\pi_K$. The attention scores can then be expressed as $Q_{\text{obf}}K_{\text{obf}}^\top = Q\pi_Q\pi_K^\top K^\top$, from which we cannot recover $QK^\top$. According to the original GroupCover paper, $Q_{\text{obf}}$ and $K_{\text{obf}}$ are sent to the TEE for recovery before computing the attention scores. The recovered results need to be encrypted using OTP before being sent to the GPU, to prevent attackers from reverse engineering $W_q$ and $W_k$. At this stage, the attention scores can be expressed as:

$$Q_{\text{OTP}}K_{\text{OTP}}^\top = (Q + P_1)(K + P_2)^\top = QK^\top + QP_2^\top + P_1K^\top + P_1P_2^\top \tag{27}$$

where $P_1$ and $P_2$ are random matrices used for OTP encryption, $Q, K, P_1, P_2 \in \mathbb{R}^{l\times d}$. In the above equation, only the term $P_1P_2^\top$ can be precomputed, while computing $QP_2^\top + P_1K^\top$ has a time complexity of $O(l^2d)$. When the input token length is too large, the computational overhead becomes unacceptable for the TEE.

For ArrowCloak, directly computing $Q_{\text{obf}}K_{\text{obf}}^\top$ is recoverable. However, some LLMs apply normalization to $Q$ and $K$ before performing the product, and this normalization must be executed within the TEE. Otherwise, $Q_{\text{obf}}K_{\text{obf}}^\top$ would become irrecoverable. In the next section, we demonstrate that OTP encryption of the RMSNorm result is necessary. Without it, the correct input could be exposed, potentially allowing adversaries to derive the weights of the preceding layer, specifically $W_q$ and $W_k$.

## E.2. Security of RMSNorm

**Recovering the RMSNorm Weight** An adversary can recover the RMSNorm weight by carefully constructing inputs. Suppose the RMSNorm weight vector $\gamma$ has dimension $d$. The adversary can choose the input vector $x = \mathbf{1}_d$, yielding

$$\text{RMSNorm}(x) = \gamma \odot \frac{\mathbf{1}_d}{\sqrt{\frac{1}{d}\sum^d 1^2 + \epsilon}} = \frac{1}{\sqrt{1+\epsilon}}\gamma. \tag{28}$$

Therefore,

$$\gamma = \sqrt{1+\epsilon}\,\text{RMSNorm}(\mathbf{1}_d). \tag{29}$$

**Recovering the Input Given the Weight and the Output** Given the output $y$, the adversary can set up the following equations:

$$y = \frac{\gamma}{k} \odot x, \qquad k = \sqrt{\frac{1}{d}\sum_{i=1}^{d}x_i^2 + \epsilon}. \tag{30}$$

This implies

$$x = \frac{k}{\gamma} \odot y, \tag{31}$$

where $\frac{k}{\gamma}$ denotes element-wise division. Substituting this expression into the definition of $k$ gives

$$k = \sqrt{\frac{1}{d}\sum_{i=1}^{d}x_i^2 + \epsilon} = \sqrt{\frac{1}{d}\sum_{i=1}^{d}\left(\frac{ky_i}{\gamma_i}\right)^2 + \epsilon}. \tag{32}$$

Solving for $k$, we obtain

$$k = \sqrt{\epsilon \Big/ \left(1 - \frac{1}{d}\sum_{i=1}^{d}\left(\frac{y_i}{\gamma_i}\right)^2\right)}. \tag{33}$$

Substituting Eq. (33) into Eq. (31) yields the recovered input.

