# OpenReview forum: "SLIM: Secure and Efficient Inference for Large Language Models on Untrusted Devices via TEEs"
_ICML.cc/2026/Conference — ICML 2026 regular_

### Official Review · Reviewer_q1Vz · 2026-03-06

**Soundness:** 3
**Presentation:** 3
**Significance:** 3
**Originality:** 3
**Overall Recommendation:** 5
**Confidence:** 3

**Summary:**

This paper presents privacy preserving inference for LLM under TEE protection. This is one of the few SOTA research that can support LLM, demonstrating a timely contribution to the broader community. To reduce the cost of dense orthogonal matrix multiplication, the authors proposed T-way mixing algorithm, drastically minimizing the computational overhead. The accuracy of inference correctness is also maintained at high level despite the strong protection provided. Experimental results show that this work can withstand commonly found statistical attacks up to satisfying level.

**Compliance With Llm Reviewing Policy:**

Affirmed.

**Final Justification:**

I'm keeping the score the same.

**Key Questions For Authors:**

1)	What is the OTP in pg. 5 (below Fig. 4) means? Is it a cryptographically secure encryption like AES/SHA3?
2)	Pg. 7, “Impact of Different Block Sizes…”, what is the block size here means? Does it refer to the Householder block size?
3)	Why is the Householder block size set to 4 in SLIM, is this related to the computational performance? What would be the effect of other block size and how to fine-tune this for various application scenarios?

**Limitations:**

There was limited discussion on the limitations of this work.

**Strengths And Weaknesses:**

Strengths:
1)	The proposed inference framework can achieve much higher accuracy compared to existing SOTA, which is a notable contribution to the community.
2)	The T-way mixing algorithm seems to be very efficient, it is the highlight of this paper.
3)	Rigorous experiments show that SLIM can resists standard statistical analysis very well compared to SOTA.
Weaknesses:
1)	The obfuscation process introduces many computational steps, which introduces noticeable errors when the inference data propagates. For example, Decoder block requires a few division operations, what is the effect on the data, does it introduces more errors?  There is no error bound analysis on this aspect.

---

> ### Author Rebuttal · Authors · 2026-03-31
>
> Thank you very much for your positive evaluation, and especially for pointing out that SLIM is a meaningful contribution to the community.
>
> **A1: On error accumulation and the lack of an error bound**
> SLIM is strictly computationally equivalent to the original model under exact real arithmetic. Its obfuscation transformation consists of permutations and Householder-based orthogonal transformations, all of which are invertible and norm-preserving. Under exact real arithmetic, the SLIM transformation is functionally equivalent to the original model. We agree, however, that the current paper does not yet provide a formal finite-precision error bound. For the extra division involving Norm weights, we replace zero elements in the implementation and ensure that the replacement values are sufficiently small so that the division remains valid and no overflow occurs.
>
> Therefore, the tiny deviations observed in actual inference mainly come from standard numerical factors at the engineering level, rather than from the SLIM mechanism itself, including:
>
> 1. rounding errors in finite-precision floating-point computation and their accumulation across layers;
> 2. some models (e.g., Gemma3-1B) contain zero elements in their scaling weights, so these values must be replaced with sufficiently small nonzero values to guarantee invertibility of the transformation.
>
> We have also added inference error experiments under different numerical precisions in our response to Reviewer pcWB to further clarify this point.
>
> **A2: On the meaning of OTP**
> The OTP (One-Time Pad) is a cryptographic encryption method. Unlike AES or SHA3, OTP uses a truly random key that is as long as the message, used exactly once, and kept strictly secret, and combines it with the plaintext bit by bit or character by character. OTP used in SLIM refers to a fresh one-time random masking pad used to protect the recovered feature before lm_head computation. We use the term in the sense of one-time masking and will clarify the implementation assumptions explicitly. In SLIM, each OTP noise mask is unique and used only once, which ensures that the attacker cannot infer the noise matrix through training or other methods.
>
> **A3: On the definition of block size**
> The block size in the paper refers to the **Householder block size $k$**, i.e., the dimension of each Householder sub-block in the block-diagonal Householder matrix.
>
> **A4: On why we choose $k=4$**
> The choice of block size essentially reflects **a trade-off between security and efficiency**. We set the Householder block size to 4 based on a joint consideration of theoretical analysis and experimental results. According to the obfuscation effectiveness analysis in Section 4.2 of the paper, the design principle of the T-Way Mixing algorithm, and the experiments in Figures 5 and 6, a smaller block size, when combined with a sufficient number of mixing rounds, more easily approaches the target random distribution and thus achieves stronger directional obfuscation. By contrast, when the block size is too large, the random Householder blocks become closer to the identity transformation, which reduces the effectiveness of each mixing step; meanwhile, too many mixing rounds increase the TEE computation overhead. We have supplemented quantitative experimental results on the impact of different block sizes on obfuscation effectiveness and TEE computation overhead, and the results are consistent with our expectation. The detailed data can be found in our response to Reviewer GiGw-A1. Therefore, block size $=4$ achieves the best practical balance among protection strength, computational cost, and deployment feasibility. **In practical deployment**, block size and mixing rounds can also be tuned jointly: for example, in latency-sensitive scenarios, one may use the optimal number of rounds, while in scenarios with a more generous compute budget and stronger security requirements, more mixing rounds can be considered for enhanced protection.

---

> > ### Author Rebuttal · Reviewer_q1Vz · 2026-04-02
> >
> > All my comments are addressed fully.

---

### Official Review · Reviewer_h5iL · 2026-03-09

**Soundness:** 3
**Presentation:** 2
**Significance:** 3
**Originality:** 2
**Overall Recommendation:** 4
**Confidence:** 3

**Summary:**

This paper aims to defend against on-device language model extraction attacks with TEEs. The threat model is that the model weights will be deployed on a local device where all information can be directly obtained. Therefore, they design an effective private inference algorithm with the help of TEE. Specifically, the victim model will be encrypted and then deployed and executed in the GPU of the local device, whereas only intermediate embeddings are required to be passed into TEEs for decryption. Experiments show that this method can significantly reduce the model extraction efficacy under a high token generation throughput.

**Compliance With Llm Reviewing Policy:**

Affirmed.

**Final Justification:**

I sincerely appreciate the detailed explanation of their study.

Now I agree with the authors of the value of this threat model. Yes, it is used to defend against some side-channel extraction attacks for local devices, aiming to degrade the threat model into black-box extraction attacks.  Therefore, the first the second concerns have been well answered. I fully acknowledge the importance of this threat model, which makes me change my score.

Thanks for your patience in rebuttal.

**Key Questions For Authors:**

Please kindly refer to weaknesses.

**Limitations:**

yes

**Strengths And Weaknesses:**

## Strengths
- The proposed method is very interesting and solid.
- The paper provided many comparisons (both performance and speed) among current baselines. The results are great.

## Weakness
- Currently, most of the model stealing attacks on LLMs are based on a black-box threat model. SLIM cannot defend against this attack. For instance, given a local victim model protected via SLIM, the attacker can still collect query-response pair datasets and use them to accomplish model extraction. Therefore, I worry about the realistic application of this study.
- Moreover, the exposure of layer-wise hidden states (e.g., $y^{(1)}$) would lead to layer-based model extraction. In constrast, many MPC strategies have considered and addressed this problem.
- There lacks adaptive attack experiments for SLIM. For instance, will the exposed embedding vectors be used to uncover the computation within TEEs and thus recover the model weights?

---

> ### Author Rebuttal · Authors · 2026-03-31
>
> Thank you very much for your careful review and for recognizing the technical idea and experimental results of this work.
>
> **A1: On black-box extraction and practical significance**
> We agree that SLIM is not intended to defend against black-box model stealing via query-response pairs. As long as a model exposes an inference interface, such attacks remain possible in principle, and this is not unique to SLIM. However, this does not diminish the practical value of our work, because SLIM addresses a different and stronger threat in deployment on untrusted devices.
>
> Our setting considers private models deployed on user-side or other untrusted devices. In this setting, an attacker can not only issue queries, but also control the host device and access the model structure, obfuscated weights, and intermediate states outside the TEE boundary. Without protection, this introduces an additional white-box attack surface, which is usually lower-cost, more effective, and more efficient than black-box stealing.
>
> Specifically, SLIM keeps the critical inverse transformation inside the TEE, so an attacker outside the TEE can only obtain an obfuscated model with degraded utility, rather than the original model or a functionally equivalent one. Thus, the goal of SLIM is to degrade white-box stealing into black-box stealing. This threat model and defense goal are consistent with prior works such as ArrowCloak (USENIX Security 2025), TSQP (IEEE S&P 2025), CoreGuard (NeurIPS 2025), and GroupCover (ICML 2024). The practical value of SLIM therefore lies in significantly reducing the high-risk white-box attack surface exposed by deployment on untrusted devices, making such deployment safer and more practical. We will also further clarify and emphasize SLIM’s practical value in the revision.
>
>
> **A2: On layer-wise hidden states and the relation to MPC**
> If a system directly exposes plaintext layer-wise hidden states to an untrusted environment, this indeed creates a layer-wise extraction risk. This is exactly what SLIM is designed to prevent. The untrusted side in SLIM does not observe plaintext layer-wise representations, but obfuscated intermediate states that remain in the transformed space throughout execution. SLIM is designed so that these transformed representations propagate across layers, thereby preventing layer-by-layer recovery and exposure. Appendix B further theoretically shows that attackers cannot recover the critical transformation solely from intermediate features.
>
> From the perspective of objective, MPC and the SLIM/TEE line are aligned: both aim to prevent intermediate-layer information from being used for model extraction. The difference lies in the mechanism. MPC uses secret sharing or encrypted-state protocols to avoid plaintext exposure and thus provides stronger cryptographic protection, but its substantial communication and interaction overhead still makes practical support for large-scale computational workloads difficult at present. TEE/SLIM instead relies on a hardware root of trust, keeps the key inverse transformation inside the TEE, and exposes only protected obfuscated representations outside. In this sense, the two directions are better viewed as complementary rather than competing, and combining them may be a promising avenue for future work.
>
> **A3: On adaptive attacks and whether exposed embeddings can recover weights**
> We thank the reviewer for this suggestion. We agree that adaptive attacks are an important direction for evaluating SLIM. However, SLIM already assumes that the attacker can access all models and intermediate data outside the TEE. Therefore, exposing embeddings does not go beyond our threat model. The key point is that although the attacker can observe this TEE-external information, they cannot access the code and data inside the TEE, nor can they uniquely recover the private transformations introduced inside the TEE or the original weights from such observations; Appendix B already analyzes this point.
>
> In addition, we have evaluated the adaptive attack closest to this concern, namely the $\Pi$-reconstruction attack, where the attacker explicitly trains a module to fit the hidden transformation after the first layer. The results show that even with stronger data collection capability, SLIM remains close to Shield-Whole and far from No-Shield. We will make this clearer in the revision and explicitly discuss adaptive attacks leveraging exposed embeddings.
>
> Reviewer pcWB also raised this issue, and we have added a joint multi-layer attack experiment (Response to Reviewer pcWB-A2). The results show that SLIM can resist such joint multi-layer attacks.
>
> Again, thank you very much for your time and effort. We hope that our clarifications above adequately address your concerns. If you feel these revisions strengthen our paper, we would welcome any upward adjustment of your review.

---

> > ### Author Rebuttal · Reviewer_h5iL · 2026-04-03
> >
> > Dear authors,
> >
> > I sincerely appreciate your response.
> >
> > Unfortunately, I still cannot accept the view that this threat model is practical. My core concerns come from this aspect.
> >
> > > Our setting considers private models deployed on user-side or other untrusted devices. In this setting, an attacker can not only issue queries, but also control the host device and access the model structure, obfuscated weights, and intermediate states outside the TEE boundary. Without protection, this introduces an additional white-box attack surface, which is usually lower-cost, more effective, and more efficient than black-box stealing.
> >
> > Can you further explain your threat model? Thanks.

---

> > > ### Author Response · Authors · 2026-04-05
> > >
> > > Dear Reviewer,
> > >
> > > We sincerely thank you for your reply and the opportunity to clarify this point. SLIM does not aim to eliminate black-box attacks entirely. Instead, it aims to remove the additional risk introduced by runtime exposure on untrusted devices. We explain this below.
> > >
> > > **First, we clarify the practical relevance of this scenario.**
> > > SLIM targets model deployment on untrusted devices, which is common in practice:
> > > 1. **Open edge devices**, where local deployment reduces latency and improves privacy, e.g., Google exposes Gemini Nano through Android AICore and ML Kit [1].
> > > 2. **Honest-but-curious cloud platforms**, where model owners deploy private models on third-party infrastructure to reduce operational costs and support elastic scaling, e.g., Anthropic provides Claude through Google Vertex AI [2].
> > >
> > > In such scenarios, the attacker is not limited to black-box extraction, but may also exploit system-level runtime exposure to recover model structure or weights. Prior works have shown weight extraction through memory side channels [3], model extraction through simple dynamic analysis [4], and leakage of model-sensitive data from GPU local memory, as highlighted by LeftoverLocals (CVE-2023-4969) [5]. Therefore, deployment on untrusted devices introduces **additional runtime risks beyond black-box access**. These attacks are often lower-cost and more effective than black-box attacks [3], [4], [8], which usually require repeated queries and substantial training costs. This is the practical motivation of SLIM.
> > >
> > > **This is also why TEE is needed.**
> > > TEE provides a strong defense against such runtime attacks [3], [6], [7]. For example, in Intel SGX, enclave pages remain encrypted in main memory and are only decrypted inside the CPU during execution. Thus, attackers cannot directly obtain meaningful plaintext model data via the above attacks. However, placing the entire LLM inside a TEE incurs prohibitive performance overhead. This motivates **TEE-based obfuscation frameworks** such as SLIM: the obfuscated model is executed on the untrusted GPU, while only the most security-critical transformations remain inside the TEE. This design minimizes TEE overhead while preserving the security of inference. Even if runtime attacks occur, the attacker can only obtain obfuscated and unusable model information.
> > >
> > > **With this clarified, we restate the threat model of SLIM.**
> > > The adversary aims to steal the functionality of the deployed model. The threat model assumes that the adversary controls the host-side software stack outside the TEE and can access all model information exposed outside the TEE, while code and data inside the TEE remain inaccessible. Possible access channels include memory scanning [3], dynamic analysis [4], installation package unpacking [4], and bus monitoring [8]. The threat model further assumes that the attacker may infer, from the information exposed outside the TEE, which public pretrained model was used as the fine-tuning base. This is a common and practical assumption in model stealing research [7], [10].
> > >
> > > The defender’s goal is to **downgrade these attacks to black-box attacks**. Black-box extraction can be viewed as the upper bound of security, since it may still occur even when the model is deployed only on the owner’s server. Thus, SLIM does not aim to eliminate black-box attacks entirely. Instead, it aims to remove the **additional security risk introduced by runtime exposure on untrusted devices**. This is achieved by combining weight obfuscation with TEE isolation. TEE-specific side-channel attacks are not considered in the current threat model. To the best of our knowledge, SLIM’s threat model is consistent with prior work [7], [9], [10].
> > >
> > > We thank you again for your careful review and valuable question. We will make these points clearer in the revision and sincerely hope this response addresses your concern.
> > > ***
> > > **References:**
> > >
> > > [1] https://developer.android.com/ai/gemini-nano
> > >
> > > [2] https://docs.cloud.google.com/vertex-ai/generative-ai/docs/partner-models/claude
> > >
> > > [3] DeepSteal: Advanced Model Extractions Leveraging Efficient Weight Stealing in Memories. IEEE S&P, 2022.
> > >
> > > [4] Mind Your Weight(s): A Large-scale Study on Insufficient Machine Learning Model Protection in Mobile Apps. USENIX Security, 2021.
> > >
> > > [5] https://blog.trailofbits.com/2024/01/16/leftoverlocals-listening-to-llm-responses-through-leaked-gpu-local-memory/
> > >
> > > [6] https://www.anthropic.com/research/confidential-inference-trusted-vms
> > >
> > > [7] Game of Arrows: On the (In-)Security of Weight Obfuscation for On-Device TEE-Shielded LLM Partition Algorithms. USENIX Security, 2025.
> > >
> > > [8] Hermes Attack: Steal DNN Models with Lossless Inference Accuracy. USENIX Security, 2021.
> > >
> > > [9] TSQP: Safeguarding Real-Time Inference for Quantization Neural Networks on Edge Devices. IEEE S&P, 2025.
> > >
> > > [10] GroupCover: A Secure, Efficient and Scalable Inference Framework for On-device Model Protection based on TEEs. ICML, 2024.

---

### Official Review · Reviewer_GiGw · 2026-03-11

**Soundness:** 3
**Presentation:** 2
**Significance:** 2
**Originality:** 3
**Overall Recommendation:** 4
**Confidence:** 3

**Summary:**

This paper addresses the critical challenge of reconciling strong security guarantees and high inference efficiency for large language models (LLMs) deployed on untrusted hardware, where weight extraction and model theft pose severe intellectual property risks. Existing TEE-based weight obfuscation schemes either suffer from prohibitive overhead due to per-layer TEE interactions or lack robust security against statistical analysis attacks. To solve this problem, the authors propose SLIM, a secure lightweight inference framework that leverages the iterative architectural nature of LLMs to enable TEE-transformed feature representations to cascade through consecutive obfuscated layers, thus minimizing TEE interactions to a single lightweight authorization per inference request. At the core of SLIM is the T-Way Mixing algorithm, which achieves thorough weight obfuscation via carefully constructed block-diagonal Householder matrices (for inter-vector covering) and successive random permutations, while reducing TEE-side computational complexity by an order of magnitude through vectorized operations. The authors conduct extensive theoretical analyses (including security proofs for obfuscation effectiveness and transformation matrix unidentifiability) and empirical evaluations on four representative LLMs (GPT2-Small, Gemma3-1B, LLaMA3.2-3B, Qwen3-4B) across diverse datasets.

**Compliance With Llm Reviewing Policy:**

Affirmed.

**Final Justification:**

My question has been answered, and my score remains unchanged taking into account the comments from other reviewers.

**Key Questions For Authors:**

1-SLIM is evaluated only on LLMs up to 4B parameters and no larger-scale models (7B/13B/70B) are tested. Larger LLMs have higher memory and computational demands, and their layer-wise feature dynamics differ from smaller models. How does SLIM scale to larger LLMs in terms of TEE memory overhead, end-to-end inference throughput, and obfuscation effectiveness? Have you conducted any preliminary experiments or theoretical analyses on larger LLMs, and are there any modifications needed for SLIM to adapt to large-scale LLM deployment?

2-SLIM’s Post-RMSNorm design uses a branch-first, merge-later paradigm with a randomly generated orthogonal matrix U for additional obfuscation. The paper states that U does not affect feature transformation correctness but strengthens obfuscation—yet there is no quantitative analysis of U’s impact on security (e.g., how U reduces the success rate of model extraction attacks) and efficiency (e.g., whether U introduces additional GPU/TEE overhead for different input lengths). What is the quantitative impact of U on SLIM’s security and efficiency, and how is the random orthogonal matrix U generated (e.g., distribution, computational cost of generation)?

3-SLIM is KV-cache friendly, a key advantage over SOTA schemes, but the paper provides no quantitative evaluation of the overhead savings from KV-cache compatibility (e.g., comparison of end-to-end inference latency/throughput between SLIM and ArrowCloak/GroupCover for long sequence generation with KV-cache enabled). What is the exact quantitative benefit of SLIM’s KV-cache compatibility for long sequence LLM inference (e.g., input/output sequences with 1024+ tokens), and does SLIM’s KV-cache design introduce any security tradeoffs (e.g., exposing obfuscated K/V vectors to the untrusted GPU)?

**Limitations:**

The authors have not adequately discussed the limitations and potential negative societal impact of their work. The paper should add a dedicated section to discuss the key limitations of SLIM, including (1) the exclusion of TEE side-channel attacks in the threat model and the resulting practical security risks; (2) the current limitation to small/medium-scale LLMs and scalability challenges for large-scale models; (3) the narrow TEE/hardware validation scope and porting challenges for edge devices; (4) the hyperparameter sensitivity of the T-Way Mixing algorithm and the lack of automated tuning.

**Strengths And Weaknesses:**

Strengths:

1-SLIM’s key insight of cascading TEE-authorized feature representations through LLM layers eliminates the per-layer TEE interaction bottleneck of existing schemes, a fundamental improvement to TEE-based LLM secure inference that directly addresses the core tradeoff between security and efficiency.
2-The T-Way Mixing algorithm intelligently combines block-diagonal Householder matrices and random permutations, balancing obfuscation effectiveness (disrupting vector directional information) and TEE computational efficiency (reducing matrix multiplication to vector operations). The theoretical derivation of the optimal block size and transformation rounds further strengthens the algorithm’s rigor.


Weaknesses:

1-The evaluated LLMs are limited to small/medium scales (up to 4B parameters); there is no analysis of SLIM’s performance, memory overhead, and obfuscation effectiveness on larger LLMs (e.g., 7B, 13B, 70B), which are the focus of real-world deployment and pose greater challenges for TEE-based secure inference.

2-While the paper recommends a block size of k=4, there is limited analysis of the sensitivity of SLIM’s security and efficiency to key hyperparameters (e.g., transformation rounds, block size) across different LLM architectures and input lengths. The paper also does not discuss the automated tuning of these hyperparameters for different LLMs.

3-Although SLIM fully implements OTP for output processing, the paper provides only high-level details of the OTP design and no quantitative evaluation of its computational and memory overhead—an important component for end-to-end inference efficiency, especially for long input sequences.

---

> ### Author Rebuttal · Authors · 2026-03-31
>
> Thank you very much for your valuable and constructive feedback.
>
> **A1: On the scalability to large models, and adaptation to large-scale deployment**
>
> To demonstrate end-to-end scalability on large models, we additionally report TTFT/throughput under different block sizes:
>
> |token length|k|Qwen3-14B|Qwen3-4B|GLM-4.7-Flash (30B)|Gemma3-1B|LLaMA3.2-3B|GPT2-Small|
> |-|-|-|-|-|-|-|-|
> |128|Pure GPU|242.4/20.7|74.8/37.9|132.8/15.8|36.0/43.2|56.3/55.6|7.6/208.3|
> ||2|332.0/16.5|133.9/28.1|186.5/14.7|81.1/31.9|121.0/36.3|48.1/93.7|
> ||4|292.8/17.5|107.4/30.8|155.3/14.8|60.0/34.2|90.1/41.7|26.8/114.3|
> ||8|278.9/17.9|100.9/31.6|145.0/14.9|50.2/35.0|83.5/42.9|22.0/120.6|
> ||16|275.2/18.1|93.9/32.2|142.6/14.9|46.6/36.0|80.8/43.5|19.4/123.6|
> ||32|270.2/18.1|93.8/32.6|142.5/15.0|43.9/36.6|75.7/43.9|16.2/127.2|
> |1024|Pure GPU|1737.9/12.1|551.4/27.9|162.9/15.6|124.0/38.5|420.1/40.0|17.3/-|
> ||2|2048.5/10.0|715.4/21.3|291.1/13.7|246.2/28.1|602.5/27.4|89.2/-|
> ||4|1936.0/10.7|644.0/23.2|221.6/14.7|202.9/30.2|511.0/30.8|55.3/-|
> ||8|1869.9/10.9|623.2/23.8|208.8/14.8|182.6/31.3|496.4/31.6|45.3/-|
> ||16|1853.0/11.0|607.5/24.2|195.7/14.8|178.5/31.3|479.2/32.5|40.4/-|
> ||32|1849.8/11.0|604.1/24.4|190.2/15.0|173.3/31.9|478.5/32.8|35.6/-|
>
> Due to the token length limit of GPT2, results for 1024 cannot be obtained.
>
>
> In addition, we quantify obfuscation effectiveness using the following KL metric:
> $KL(\cos(W_{\mathrm{obf}},W_{\mathrm{pre}}),\cos(W_{\mathrm{random}},W_{\mathrm{pre}}))$.
>
> |Model|k=2|k=4|k=8|k=16|k=32|
> |-|-|-|-|-|-|
> |Qwen3-14B|6.7|9.5|25.1|32.0|32.9|
> |GLM-4.7-Flash (30B)|3.6|8.5|24.2|31.2|32.0|
> |Gemma3-1B|2.5|7.1|24.2|30.6|31.6|
> |LLaMA3.2-3B|2.4|8.5|23.0|31.6|32.4|
> |Qwen3-4B|3.8|9.7|24.7|31.3|32.2|
> |GPT2-Small|0.6|1.6|15.7|30.3|32.3|
>
> These results show that as $k$ increases, obfuscation effectiveness decreases while inference performance improves. Based on the elbow principle of the Pareto curve, we choose $k=4$ as the trade-off between security and efficiency.
>
> **A2: On OTP overhead**
>
> TEE memory mainly consists of two parts: the obfuscation matrices and the OTPs. For the obfuscation matrices, we store their **factorized form** inside the TEE, so the memory cost is $O(2d\lceil\log_kd+\log_k(\ln\frac{d^{2}}{1-p})\rceil)$. The exact OTP memory cost depends on the pad-buffering policy. In our current prototype, we preload a **fixed-size buffer of fresh random pads** and the corresponding `lm_head(Pad)` terms into secure memory. We do not claim that this prototype alone constitutes a production-ready OTP pipeline; real deployment would require explicit TEE-side pad lifecycle management, which we will state as a limitation / future engineering direction.
>
>
> **A3: On the security and efficiency impact of the random orthogonal matrix $U$**
>
> We generate $U$ by applying QR decomposition to a standard Gaussian random matrix, whose orthogonal factor follows the Haar distribution. This process is performed **before model deployment**, and its generation cost is very small, so it does not affect inference efficiency.
> From the security perspective, $U$ disrupts directional information. Our experiments show that after the $U$ transformation, the cosine distribution between the transformed weights and the pretrained weights becomes highly similar to the distribution of “random vectors vs. pretrained weights,” with a KL divergence **below 0.1**.
> From the efficiency perspective, the two Linear layers after the Post-RMSNorm branch can be executed in parallel using different CUDA streams. In our experiments, compared with the case without the additional branch, the branch-merge structure has **no measurable impact on either TTFT or throughput**.
>
> **A4: On the benefit-security trade-off of KV-cache**
>
> As shown in Appendix A, once ArrowCloak (AC) and GroupCover (GC) enable per-layer OTP, they become difficult to reconcile with KV-cache. Therefore, in the paper experiments, we did not enable per-layer OTP for AC/GC; instead, we enabled KV-cache for SLIM, AC, and GC alike, so as to avoid conflating cache compatibility with other runtime differences. It should be noted that this setting actually **favors the performance of AC/GC**, but at the same time weakens their security, because they expose both pre-obfuscation and post-obfuscation KV vectors. In contrast, SLIM exposes only the correct KV required for inference. Since the attacker does not possess the paired obfuscated KV needed to recover the transformation, the same risk does not arise for SLIM (see Appendix B.2).
>
> Quantitatively, KV-cache mainly improves efficiency in the decoding stage:
>
> * at **token length = 1024**, throughput with KV-cache is about **14×** that without KV-cache;
> * at **token length = 128**, the gain is about **4×**.
>
> We will further clarify these benefits and their security implications in the revision.

---

### Official Review · Reviewer_pcWB · 2026-03-11

**Soundness:** 3
**Presentation:** 3
**Significance:** 3
**Originality:** 2
**Overall Recommendation:** 4
**Confidence:** 3

**Summary:**

This paper studies TEE-based protection for edge-deployed LLMs and proposes **SLIM**, whose core idea is to perform a single TEE authorization after the first decoder block and then keep the hidden representation in a transformed space so that later obfuscated blocks preserve authorization automatically. To make this practical, the paper introduces T-Way Mixing, which combines block-diagonal Householder transforms with repeated permutations so the TEE only performs lightweight matrix-vector work while the obfuscation still hides vector directions. Empirically, the paper evaluates GPT2-Small, Gemma3-1B, LLaMA3.2-3B, and Qwen3-4B, and reports up to **13.80x** TTFT reduction over prior obfuscation baselines together with **100% Top-1 token agreement in FP32**.

**Compliance With Llm Reviewing Policy:**

Affirmed.

**Final Justification:**

My final recommendation remains **weak accept**. I think the paper has a strong core idea, careful technical development, and meaningful empirical gains on an important problem, but the contribution feels more like a strong refinement/synthesis than a fully new paradigm, and some claims still need to be scoped carefully. The rebuttal was helpful and addressed several of my questions, but it did not materially change my overall evaluation, since the main remaining concerns are about core aspects of the work, especially the scope of the guarantee and the breadth of the evaluation.

**Key Questions For Authors:**

1. What is the precise delta over CoreGuard's propagation protocol(NIPS'25)? Is the main novelty the combination of propagation with direction-hiding orthogonal mixing, rather than one-time authorization itself?

2. Have you tried a **joint multi-layer attack** that uses all transformed layers together to estimate a functionally equivalent global transform, rather than attacking the first block or per-layer statistics separately?

3. For shared `lm_head`, what exact tensors cross the TEE boundary during decoding? What is the true asymptotic and measured cost once output processing is fully accounted for? Are **sampling**, **log-prob outputs**, and **beam search** supported exactly, or is the guarantee mainly top-1 / argmax equivalence?

4. When the paper says `lm_head(OTP)` can be precomputed, what exactly is precomputed? Is the OTP basis/pad **fresh per request/token**, or is any secret basis reused across queries? Please clarify this carefully given recent static-basis attacks.

5. Can the authors add at least one of the following: BF16/FP16/INT4 exactness, a MoE/GQA/MLA model, or an ARM TrustZone-style platform? Right now the evidence is strongest for dense FP32 models on SGX+A100, not for the full claimed deployment space.

**Limitations:**

No. The paper includes only a brief generic impact statement and does not substantively discuss limitations or possible negative societal impact. I would encourage the authors to explicitly acknowledge that: the threat model excludes TEE-specific side-channel attacks; the empirical evaluation is limited to four dense models, FP32 precision, and an SGX+A100 setup; and the strongest exactness evidence is Top-1 agreement in FP32 rather than a broader analysis of all decoding modes and deployment settings. On societal impact, the paper should discuss the dual-use risk that stronger model-theft resistance can also make external auditing, red-teaming, and accountability harder for proprietary deployed models, and should briefly state recommended safeguards and scope boundaries.

**Strengths And Weaknesses:**

**Strengths**

* The paper tackles an important and timely problem: how to protect valuable LLM weights on untrusted devices without paying the prohibitive cost of running the full model inside the TEE. The high-level design is clear and nontrivial: rather than recovering every layer inside the enclave, SLIM turns authorization into a propagating transformed hidden state that cascades through later decoder blocks.

* The strongest technical contribution is the "algebraic fit to transformer structure". The paper does real work to show how RMSNorm/LayerNorm, attention projections, residual paths, and the MLP can be rewritten so that the GPU computes correct outputs while the representation remains in the transformed space. This is much stronger than a vague "we can obfuscate transformers" claim, and the derivation around the obfuscated decoder block is one of the best parts of the paper.

* The T-Way Mixing design is well motivated. The paper identifies that a single large Householder block is a poor obfuscator because it leaves cosine similarity too concentrated, and it studies the tradeoff between block size, number of rounds, and security. I found this part insightful: it is not just a heuristic, but an attempt to explain why small blocks plus repeated mixing are the right compromise.

* The empirical results against the paper’s chosen statistical-analysis threat model are strong. The paper explicitly evaluates against the direction-similarity attack line exemplified by ArrowMatch/ArrowCloak, which is the right baseline family to beat here, and the attack-defense curves suggest SLIM is substantially harder to recover than earlier lightweight obfuscation schemes.

* The efficiency evaluation is also compelling within the compared set. SLIM reports much lower transfer/computation overhead than GroupCover and ArrowCloak, and the authors make the comparison "conservative" by omitting OTP overhead for those baselines while fully implementing OTP for SLIM. That is a fair experimental choice in the paper’s favor.

**Weaknesses**

* The paper **overstates novelty** if it presents "one-time authorization" as its main conceptual advance. CoreGuard already introduced a **propagation protocol** that reduces TEE authorization to a **single initial authorization** for the whole model. In my view, the real novelty of SLIM is narrower and more defensible: it combines **propagation-style efficiency** with **direction-hiding orthogonal mixing** that is better aligned with recent attacks. That is still a contribution, but it should be framed more precisely.

* The **security theory is limited**. What is shown is closer to non-uniqueness / underdetermination arguments than to a cryptographic security proof. More importantly, the paper does not really test the most natural adaptive attack suggested by its own design: a **joint multi-layer recovery attack** that exploits the fact that the same transform family is propagated through many layers and many queries. The current attack suite is good, but not obviously exhaustive relative to the structure SLIM exposes.

* The **output-head / OTP path is underspecified**. The paper says the backbone output is recovered in the TEE, random noise is added, the GPU computes the prediction head, and the TEE removes the noise effect, returning only predicted indices; it also states that `lm_head(OTP)` can be precomputed. This seems fine for greedy decoding, but the paper does not clearly explain the exact decode-time data movement, the cost accounting for shared `lm_head`, or whether sampling/log-probs/beam search are supported exactly. This matters even more in light of very recent work showing that **precomputed static secret bases** can create key-reuse-style vulnerabilities in partial TEE-shielded inference. I am **not** claiming this breaks SLIM, only that the implementation details need to be spelled out carefully.

* The empirical coverage is narrower than the claims. The paper analytically argues support for MoE, GQA/MQA/MLA, RoPE, QK-Norm, gated attention, and KV-cache, but the actual experiments are limited to **four dense models up to 4B**, **FP32 only**, and an **SGX + A100** server platform. Given how much modern LLM serving relies on BF16/FP16/INT4 and on architectures beyond dense decoder-only models, I do not think the current evaluation fully supports the breadth of the compatibility claims. The platform evidence also looks narrower when compared with LoRO's SGX + TrustZone evaluation.

---

> ### Author Rebuttal · Authors · 2026-03-31
>
> Thank you very much for your positive and insightful feedback.
>
> **A1: On the difference from CoreGuard**
>
> We agree that SLIM’s key contribution is not “one-time authorization” itself, but a lightweight orthogonal obfuscation mechanism that propagates across consecutive Decoder Blocks, achieving both low TEE overhead and stronger direction hiding.
>
> **A2: On the joint multi-layer attack**
>
> We have added a joint multi-layer attack, in which the attacker trains two linear recovery modules: one for the first-block `down_proj` path and one shared across all subsequent transformed Decoder Blocks to approximate the inverse obfuscation transform. The obfuscated backbone is frozen, and only these two layers are optimized using 1%–60% constructed data. The results show that this stronger attack is **still weaker than the $\Pi$-reconstruction attack in the main paper**:
>
> |Ratio|0.01|0.1|0.2|0.3|0.4|0.5|0.6|
> |-|-|-|-|-|-|-|-|
> |Qwen3-4B/GoEmotions|0.25|0.27|0.30|0.38|0.39|0.49|0.48|
> |LLaMA3.2-3B/FinQA|0|0|0|0|0|0|0|
>
> **A3: On shared `lm_head`: cross-boundary tensors, real cost, and supported outputs**
>
> For shared `lm_head` decoding, GPU computation is $O(nld^2)$ and TEE authorization is $O(d\log_k d)$. During output processing, the TEE receives the normalized feature $y_{\mathrm{norm}}^{(n)}$ and the `lm_head` output, returns the OTP-added feature $y_{OTP}$, and finally outputs the token index. OTP addition costs $O(d)$. OTP removal and index computation cost $O(V)$, where $V$ is the vocabulary size. Measured results show that output processing accounts for only 3.70%–11.45% of total TEE decoding computation.
>
> At present, our strongest strict guarantee is **Top-1/argmax equivalence**.
> - **Sampling**: Exact sampling can be supported by extending the current Top-1 logic to Top-$K$, requiring only an additional heap-based Top-$K$ selection step inside the TEE.
> - **Beam search**: can be supported, but requires maintaining beam scores inside the TEE.
> - **log-prob**: **not supported**, because it would directly expose logits and create a new leakage surface.
>
> If implemented inside the TEE, the TEE complexity of sampling and beam search would be $O(V\log K)$ and $O(BV\log B)$, respectively, where $K$ is Top-$K$ and $B$ is the beam width.
>
> **A4: On precomputing `lm_head(OTP)` and pad reuse**
>
> What is precomputed is the per-pad pair (Pad,lm_head(Pad)). In our current prototype, we generate a fixed-size buffer of fresh random pads before inference, precompute the corresponding lm_head(Pad) terms, and load this pool into secure memory. Each generated token consumes one fresh pad, and pads are not reused across tokens or queries.
>
> We agree that this prototype should not be interpreted as a fully resolved production pipeline for pad lifecycle management. A real deployment would require an explicit pad-management mechanism inside the TEE, e.g., a buffer pool with background replenishment and admission control. Such a design is feasible for bursty workloads and low-to-moderate sustained demand, but under prolonged peak-rate decoding the pad pool could be exhausted if replenishment lags consumption. We will clarify this implementation assumption explicitly and list production-grade OTP management as a limitation / future engineering direction.
>
> **A5: On BF16/FP16, larger models, GQA/MoE/MLA, and TrustZone**
>
> |Model|FP32($M_{vic}$)-FP16($M_{SLIM}$)|FP16-FP16|FP32-BF16|BF16-BF16|
> |-|-|-|-|-|
> |Qwen3-4B|99.5|99.5|97.0|96.5|
> |LLaMA3.2-3B|99.0|99.3|94.2|95.3|
> |Gemma3-1B|95.0|95.2|73.6|73.2|
> |GPT2-Small|97.7|96.7|79.5|74.5|
> |Qwen3-14B|99.6|99.6|96.9|96.9|
> |GLM-4.7-Flash (30B)|-|87.8|-|55.1|
>
> The table reports Top-1 token accuracy; FP32-FP16 means the match rate between $M_{SLIM}$ in FP16 and $M_{vic}$ in FP32. Due to GPU memory limits, we do not yet have FP32 results for GLM-4.7-Flash (30B).
>
> Accuracy decreases under half precision, but SLIM still significantly outperforms existing baselines. We also observe that GroupCover reaches about 71% on GPT2+FP16, while most other baselines remain below 3%; GroupCover is also more prone to numerical overflow under FP16.
>
> Regarding model coverage:
> - **GQA**: Already covered by Qwen3.
> - **MoE/MLA**: We additionally include GLM-4.7-Flash (30B), which contains both MoE and MLA architectures.
> - **INT4**: Only TSQP supports quantization among mainstream methods, and SLIM does not support INT4 either.
> - **TrustZone**: We currently lack fair TrustZone experiments and will list this as future work.
>
> **Limitations and Impact Statement**
>
> We will also add dedicated Limitations and an Impact Statement in the final version, explicitly noting that the threat model excludes TEE side-channel attacks, that an engineering-ready OTP pipeline is still missing, and that broader support for low-precision / quantized formats remains future work. We will also clarify that SLIM should be paired with controlled audit interfaces and governance mechanisms in real deployment.

---

> > ### Author Rebuttal · Reviewer_pcWB · 2026-04-02
> >
> > The rebuttal addresses several of my questions and improves the paper’s clarity, but it does not materially change my overall assessment of the work’s contribution or empirical scope, so I retain my score.

---

### Decision · Program_Chairs · 2026-04-30

**Decision:**

Accept (regular)

**Comment:**

This paper studies how to serve models confidentially, i.e., protecting model weights from hardware service providers.

The proposed approach, SLIM, uses a T-Way mixing algorithm that uses repeated permutations and block-diagonal Householder transforms. This improves efficiency by designing for optimized operations on the TEE.

Overall, there was agreement from the reviewers that this work

1. tackled a timely and important problem,

2. had an interesting, novel, and sound  algorithm

3. had sound and sufficient theory

4. had sufficiently thorough empirical analysis showing significant speedups

and should be accepted to ICML.

There are some areas for improvement and requested/proposed changes that should be included in the final revision.

1. adjust claims to better reflect the scope of the contribution and evaluation.

2. better address the technical limitations of the work, including the specific limitations requested by Reviewer pcWB/Reviewer GiGw

3. better address societal implications

In addition to including the details clarified in the rebuttal into the final version.